



# Effects of subsurface water infiltration systems on land movement dynamics in Dutch peat meadows

Sanneke van Asselen[1], Gilles Erkens[1], Christian Fritz[2], Rudi Hessel[3], Jan J. H. van den Akker[3]

[1]Deltares Research Institute, Utrecht, 3584 BK, The Netherlands
[2]Department of Aquatic Ecology and Environmental Biology, Radboud University, Nijmegen, 6525 AJ, The Netherlands
[3]Wageningen Environmental Research, Wageningen, 6708 PB, The Netherlands

*Correspondence to*: Sanneke van Asselen (sanneke.vanasselen@deltares.nl)

**Abstract.** Large-scale drainage and cultivation of peat soils over the last centuries, occurring worldwide, has resulted in substantial $CO_2$ emission and land subsidence caused by peat decomposition by microbial activity, shrinkage and soil compaction. In addition, seasonal reversible vertical soil movement is caused by shrink and swell in the unsaturated zone and by poroelastic deformation in the saturated zone. To reduce $CO_2$ emission and land subsidence in drained peat soils, subsurface water infiltration systems (WIS) are expected to be a suitable measure. In this study, effects of WIS on seasonal vertical soil movements are evaluated, based on field measurements from five locations in Dutch peat meadows, for the years 2021 and 2022. For one of these locations, a 4-years timeseries was available, allowing to make a first estimate of the rate of multi-year land subsidence. At each study location, vertical soil movement has been measured using spirit levelling and extensometers, both in a parcel with a WIS and in a nearby reference parcel without any measure. Phreatic groundwater level fluctuations are found to induce soil volume decreases and increases in both the saturated and the unsaturated zone, which cause vertical land movement dynamics of up to 10 cm in the dry summer of 2022 at a location with a relatively thick (6 m) peat layer. Poroelastic deformation processes in the deeper saturated soil contribute substantially to surface level movement. In peat meadows, subsurface water infiltration systems, if correctly applied, reduce seasonal vertical soil movements while (potentially) reducing soils' resilience to drought-induced volume losses. Seasonal vertical soil surface dynamics are about an order of magnitude higher than longer term (years to decades) land subsidence rates, which are commonly in the order of mm yr$^{-1}$ in the Dutch drained peat areas. Therefore, multi-year data series are needed to filter out variations in seasonal dynamics, which are mainly introduced by annual variations in weather conditions, and more accurately estimate land subsidence.

## 1 Introduction

Many coastal plains worldwide contain abundant peat in the subsurface that has formed over millennia under conditions of decelerating Holocene sea-level rise (e.g., Stanley and Warne, 1994; Törnqvist et al., 2008; Drexler et al., 2009; Van Asselen, 2011). At present, many coastal areas are densely populated (e.g., Neumann et al., 2015). Human activities, in particular drainage for land reclamation, have resulted in large-scale ending of peat formation in peat-rich coastal plains worldwide.



30 In the Dutch coastal peatlands, for example, drainage using ditches and artificial lowering of the phreatic groundwater table in peat soils already started about 1000 years ago, to make the land suitable for agriculture. This resulted in increased peat decomposition by biogeochemical (oxidation) processes, resulting in $CO_2$ emission and land subsidence due to loss of soil volume (Erkens et al., 2016) and an increase of bulk density due to loss of the fibric structure by humification. Lowered phreatic groundwater levels may also lead to land subsidence by irreversible soil shrinkage above, and soil compaction below

35 the phreatic groundwater level. Shrinkage is caused by suction forces that develop when the soil dries due to drainage and evapotranspiration and water extraction by roots. Soil compaction is induced by loading and/or structurally lowered pore water pressure following drainage, which increases the effective stress (Terzaghi, 1943; Van Asselen et al., 2009 and references therein). In response to land subsidence following drainage, ditchwater levels and therewith phreatic groundwater levels were lowered repeatedly in historical times, to prevent the soil from becoming too wet for agricultural activities. This caused

40 additional and (still) ongoing land subsidence. Despite drainage, most reclaimed peatlands have become too wet for crop growth and are therefore at present used as pastures for dairy farming.

 Land subsidence due to peat compaction and decomposition is a slow and complex process taking place over long periods (many years, e.g., Van Asselen et al., 2009). On shorter (days to months) timescales, reversible soil vertical movements

45 commonly take place. In the unsaturated soil zone, these movements are especially caused by shrinkage and swelling of the soil as a result of periodic changes in soil suction forces driven by annual variations in evapotranspiration and precipitation. In the saturated soil, periodic variations in pore water pressures may cause seasonal poroelastic deformation causing vertical soil movements. At one of the study sites of this research, preliminary results indicated short-term vertical surface level movements of up to ~4 cm (Van Asselen et al., 2020). Irreversible land subsidence may also be affected by changes in rainfall patterns,

50 which can significantly impact the water table levels within peatlands. Changes in temperature may affect the rate of peat decomposition. Warmer temperatures can accelerate the decomposition process, leading to a loss of peat's ability to support itself and contribute to subsidence. Conversely, cooler temperatures can slow down decomposition, potentially stabilizing the peatland temporarily. The growth and decay of vegetation in peatlands can also influence subsidence. Healthy, well-established vegetation can help stabilize the peatland by providing a physical barrier against water movement and preventing the peat from

55 drying out. However, if the vegetation is damaged or removed, the peatland may become more susceptible to subsidence. These factors highlight the importance of managing peatlands to mitigate the risk of subsidence. This includes maintaining healthy vegetation, managing water levels effectively, and implementing sustainable land use practices to protect peatlands from further degradation (GPA, 2023).

60 In times of climate change, it is urgent to stop or at least reduce $CO_2$ emission from drained peat soils into the atmosphere. In addition, it is vital to stop or reduce land subsidence due to peat decomposition, irreversible shrinkage, and compaction processes in drained peatlands. Land subsidence in low-lying coastal areas may significantly contribute to relative sea level rise and increase the risk and impact of flooding (e.g., Ericson et al., 2006; Syvitski, 2008; Nicholls et al., 2021). Land





subsidence also leads to (high costs related to) damage to buildings and infrastructure and may cause salinization (e.g., Van

de Born et al., 2016). Also, short-term vertical movements may lead to damage and increased flood risks, and peat decomposition during summer periods with low groundwater levels to $CO_2$ emissions. Yet it is needed to quantify seasonal vertical movements and land subsidence in different environmental settings, determine the relative contribution of different processes to total deformations and assess the proportions of vertical movements that are reversible and irreversible. This will lead to increased understanding of soil deformation processes, which is needed to be able to take effective measures to reduce

land subsidence and $CO_2$ emission in drains peat meadows.

Peat decomposition and compaction may be reduced by rewetting peat soils; increased anaerobic conditions reduce microbiological activity, while higher pore water pressures in more saturated soils reduce the effective stress thereby reducing or halting compaction processes. Cultivated peat soils could be completely rewetted and transformed into natural peatland,

thereby losing their agricultural function. To maintain the agricultural land use function, which is often desirable for economic and cultural reasons, excessive lowering of the phreatic groundwater level in summer may be limited by installing subsurface water infiltration systems (WIS; Van den Akker et al., 2010; Querner et al., 2012). Drainpipes are brought into the soil, at a depth of about 20 cm below ditch water level, which stimulates water infiltration from the ditch into the peat soils in dry periods with high evapotranspiration. To actively maintain and steer the phreatic groundwater level at a desired depth,

independent of the ditch water level, drains may be connected to a reservoir with a pumping system (in this paper referred to as an active WIS, i.e., AWIS). Systems that are directly connected to a ditch, without a reservoir, are referred to as a passive WIS (PWIS).

The main aim of this paper is to assess effects of WIS on seasonal vertical land movement in drained peat soils, based on

analysis of elevation changes obtained by spirit levelling and extensometers, groundwater level measurements, and subsurface lithological composition. In this study we focus on seasonal vertical soil movements since the time series are yet too short to make reliable estimates of longer term (multiyear) land subsidence, although we can also make first (hypothetical) assessments of effects of WIS on longer term land subsidence. The research is part of the Netherlands Research Programme on Greenhouse Gas Dynamics in Peatlands and Organic Soils (NOBV). Data was collected in the period mid-2020 to 2022 from five study

sites distributed over the Dutch peatland area. For one of these locations data from 2019 was also available. Field measurements are vital to monitor the efficiency and applicability of WIS under different environmental circumstances, and possibly optimize these systems in order to stop or reduce land subsidence and $CO_2$ emission from drained peat soils.

## 2 Methods and study sites description

The five study sites are presented in Fig. 1. At these locations, one field plot is situated in a WIS parcel and another field plot

is situated in a nearby reference parcel without such a system, but with similar environmental conditions.





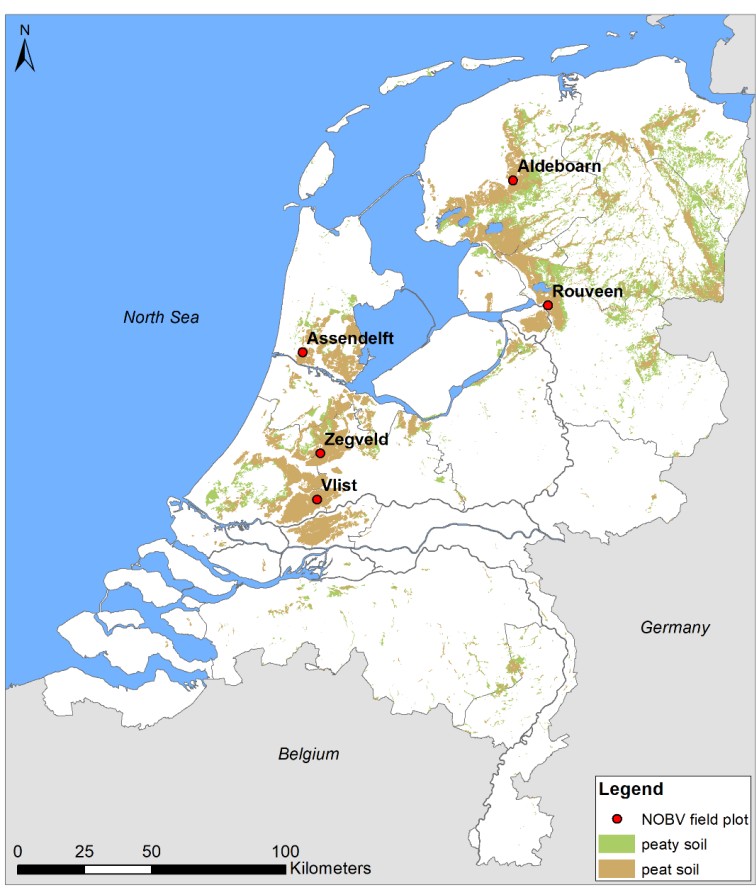

**Figure 1. Map of organic soils in the Netherlands and locations of paired field plots, each consisting of a WIS parcel and a nearby reference parcel. Source topographical map: TNO (2019). Peat and peaty soil map is derived from Brouwer et al. (2021).**

At each study location, elevation changes have been monitored in both the WIS and reference parcel using two methods: spirit

levelling and extensometry (see sections 2.1 and 2.2 respectively). In addition, groundwater and ditch water levels are monitored at each location (Van Asselen et al., 2023). Also, the subsurface has been investigated by hand corings using Edelman and gouge augers, and, at all locations except for Rouveen, also by a Cone Penetration Test (CPT; Erkens et al., 2019). In general, the subsurface at all study sites consists of Holocene peat and clay layers, overlying Pleistocene sandy deposits. Details of the groundwater system and subsurface composition at each study location are described in section 2.3. In

this study, estimates of the organic matter content mentioned in the results sections are derived from field observations from experienced geologists and soil scientists and from Loss On Ignition laboratory tests (LOI = ((dry weight − ashed weight) / dry weight) ∗ 100%); cf. Heiri et al., 2001).



## 2.1 Spirit levelling

Spirit levelling is performed four times a year along fixed transects perpendicular to and in the longitudinal direction of a
parcel, relative to a local benchmark (a steel tube) that has been founded in firm Pleistocene sand underlying the soft Holocene
peat and clay sequence. The elevation of this benchmark, relative to the Dutch Ordnance Datum (NAP), has been measured
by spirit levelling to a nearby official NAP benchmark, which are regularly checked for stability. At each location, three to
five transects of several tens of meters in length are measured per parcel, commonly using a measuring interval of 2 meters.
Measurements in ditches and on ditch banks are excluded from analyses in this study. The locations (XY coordinates) of the
begin and endpoints of the transects are determined each field campaign using an RTK-GNSS device to ensure levelling
measurements are performed at the same locations. In Aldeboarn, the distance between two measurement points ranges from
2 to 7.5 m and measuring points were marked belowground to allow for replication in time. The configuration of transects at
the five study areas is presented in section 2.3. Surface elevation is measured in winter (January), spring (April), summer (July)
and autumn (October).

## 120 2.2 Extensometry

The extensometers used in this study are specifically designed for high-resolution measurement of vertical movement of
multiple subsurface levels in soft peat and clay soils. At each of the five study sites, an extensometer has been installed in both
the WIS and the reference parcel. Each extensometer measures the vertical displacement at five anchor levels, relative to a
reference (massive cone) anchor fixed in a firm and stable sandy layer below the soft soil sequence (Fig. 2). The reference
anchor is assumed to be stable. The depth of the firm sand has at most locations been determined based on the results of the
CPT, which shows at what depth the resistance and friction of the soil significantly increases, indicating the occurrence of firm
sand. In Rouveen, the reference anchor has been installed by driving the (massive cone) anchor into the soil using a manual
pile driver. By calculating the deformation of the soil layer between two anchors, based on their vertical displacements, the
contribution of different soil layers to total surface deformation in time may be determined (Fig. 2).





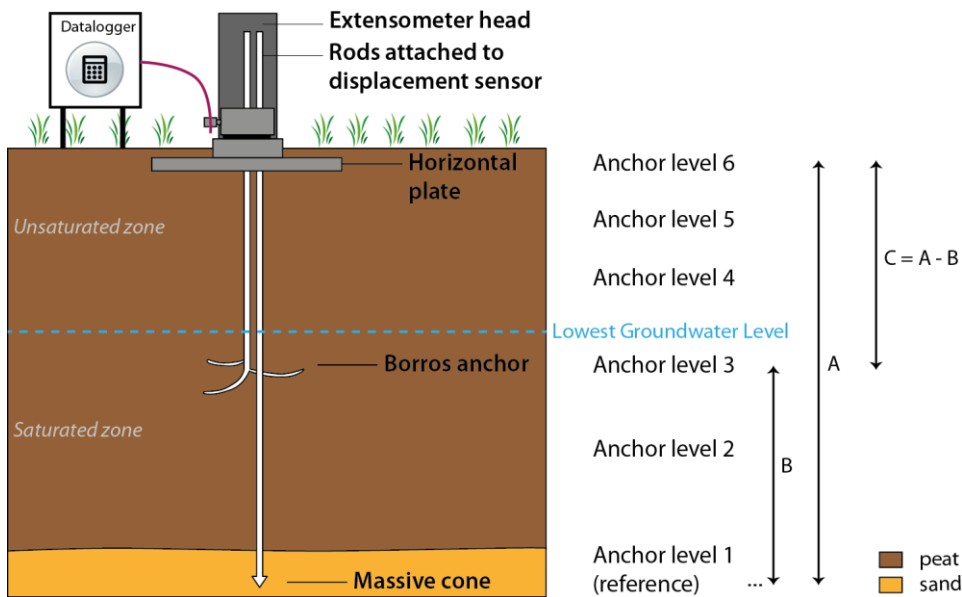


**Figure 2. Schematic representation of an extensometer set-up. Deformation of soil layers may be calculated based on elevation changes of individual anchor levels, e.g., in the example above, the deformation of layer C = elevation change of anchor level 6 (=A; relative to the reference anchor 1) – elevation change of anchor level 3 (=B; relative to the reference anchor 1).**

Different types of anchors are used (Fig. 2):

•      Anchor level 1 is a massive cone, this is the reference level. The anchor is installed using a cone penetration test vehicle. The bottom of the borehole that was made to install this anchor has been filled with a grout. This was done to prevent a connection is created between the groundwater in the Holocene and the Pleistocene sequences, which may have different hydraulic heads.

    •      Anchor levels 2 to 4 are Borros type anchors, consisting of 3 prongs that are firmly embedded in the soft soil.

140         First, a small borehole is drilled using a gouge auger. Next, the Borros anchor is brought to the desired depth, after which the prongs are extended using hydraulic pressure. Fully extended, the prongs extend circa 150 mm into the soil.

    •      Anchor level 5 is installed at a depth of circa 40 cm below surface, which is too shallow for a Borros anchor. Instead, a horizontal steel strip is used as anchor, which is brought into the undisturbed ground via a small trench.

•      Anchor level 6 is a square perforated stainless-steel plate (0.5 x 0.5 m, 8 mm perforation, about 40% open area) that is dug into the soil at a depth of circa 5 cm, thereby largely representing surface level vertical dynamics. The plate is installed at ca 5 cm depth to prevent measuring vertical movement caused by especially vegetation growth.

All anchors are connected to the extensometer head, which is fixed to the perforated stainless-steel plate (anchor level 6), with screwed stainless-steel rods (Fig. 3). The rods are protected by a ribbed tube, limiting friction of the (deforming) soil on the





rods. In the extensometer head each rod is connected to the displacement sensor, which are either vibrating wire displacement transducers or linear potentiometers. Because of limited available space in the extensometer head, this setup has a maximum of five anchor levels. The displacement sensors are connected to a datalogger next to the extensometer (Fig. 3). In this study,

measurements are recorded hourly. In the WIS parcel, the extensometer is installed halfway between two drains, in the reference parcel the extensometer is installed at about one third of the parcel width.

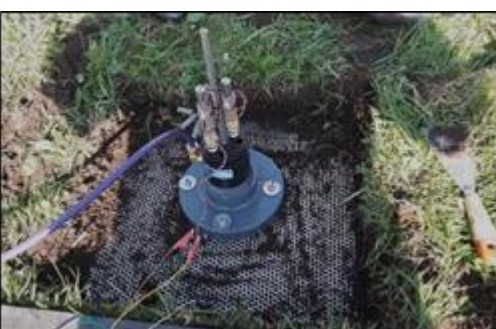
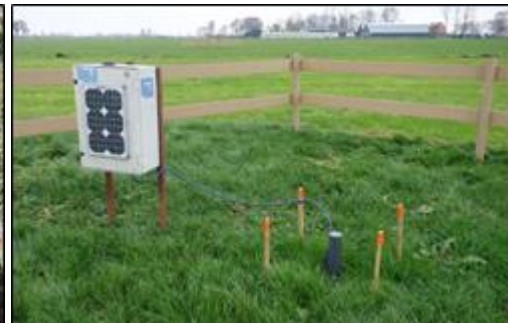

**Figure 3. Left: extensometer head and stainless-steel plate a few centimetres below surface level. Right: extensometer protected by**
**a PVC cover and connected to the datalogger with solar panels.**

### 2.3 Study sites

#### 2.3.1 Aldeboarn

In Aldeboarn, a PWIS has been installed in 2016. In the summer following the installation of PWIS, ditch water levels were raised from 80 cm to about 40 cm below surface level of the field site (Weideveld et al. 2021; Fritz et al. 2021). Vertical

elevation (changes) has been measured since using levelling and extensometry since mid-2020 (Figure 4). Drains are installed between 70 to 80 cm depth, in the longitudinal direction of the parcel. The distance between drains is ~6 m. In Aldeboarn, a HAKLAM (in Dutch: '*Hoog Als het Kan, Laag Als het Moet*', translating into 'high if possible, low if necessary') water level management has been applied. This means that the polder water level (and therefore the ditch water level of the reference plot) is set high (~45 cm below surface level) during dry periods and low if necessary for agricultural practice, such as for early

growing season fertilization and/or the harvest of the last grass cut, which require sufficient load-bearing capacity of the peat soil. The ditch water level in the reference parcel has fluctuated between ~45 cm below surface level in the summer of 2020 and ~90 cm below surface level in the first months of 2022 (Van Asselen et al., 2023). In the PWIS parcel, the ditch water level is fixed at ~45 cm below surface level since the growing season of 2022. Before that time, the ditch water level fluctuated between ~40 cm and ~80 cm below surface level (Van Asselen et al., 2023). Unfortunately, a reliable measurement of the

hydraulic head in the sand layer below the peat layer is lacking at Aldeboarn, but the hydrological model LHM4.1 (LHM, 2023) indicates that on average this site is neutral (no significant seepage or infiltration). Further information about the groundwater measurements and the PWIS system, and how it is connected to the ditches, is described by Van Asselen et al. (2023).





The subsurface in Aldeboarn is generally characterized by clay on peat on detritus on sand (see also soil profiles at extensometer locations in Table 1 and Erkens et al., 2019). The top clay layer is 0.25 to 0.65 m thick, deposited in a marine environment. The marine clay is often stiff and has an organic matter content (i.e., LOI) of about 15%. Peat fragments may occur in the clay layer due to ploughing (Fritz et al., 2021). Below the clay layer, to a maximum depth of 2.1 m below surface, the subsurface consists of oligotrophic peat, containing remains of *Sphagnum* mosses, *Eriophorum* and heather. The drained

peat is strongly decomposed and amorphous, usually down to 65 cm depth, and up to a depth of at maximum 0.95 m. At the transition to underlying Pleistocene sand, an amorphous sandy detritus layer of 0.05 to 0.20 m thick is present. The top few meters of the Pleistocene deposits consist of an alteration of sandy and clayey layers. Firm sand, in which the reference anchor of the extensometer is installed, is encountered at a depth of about 7 to 8 m below surface.

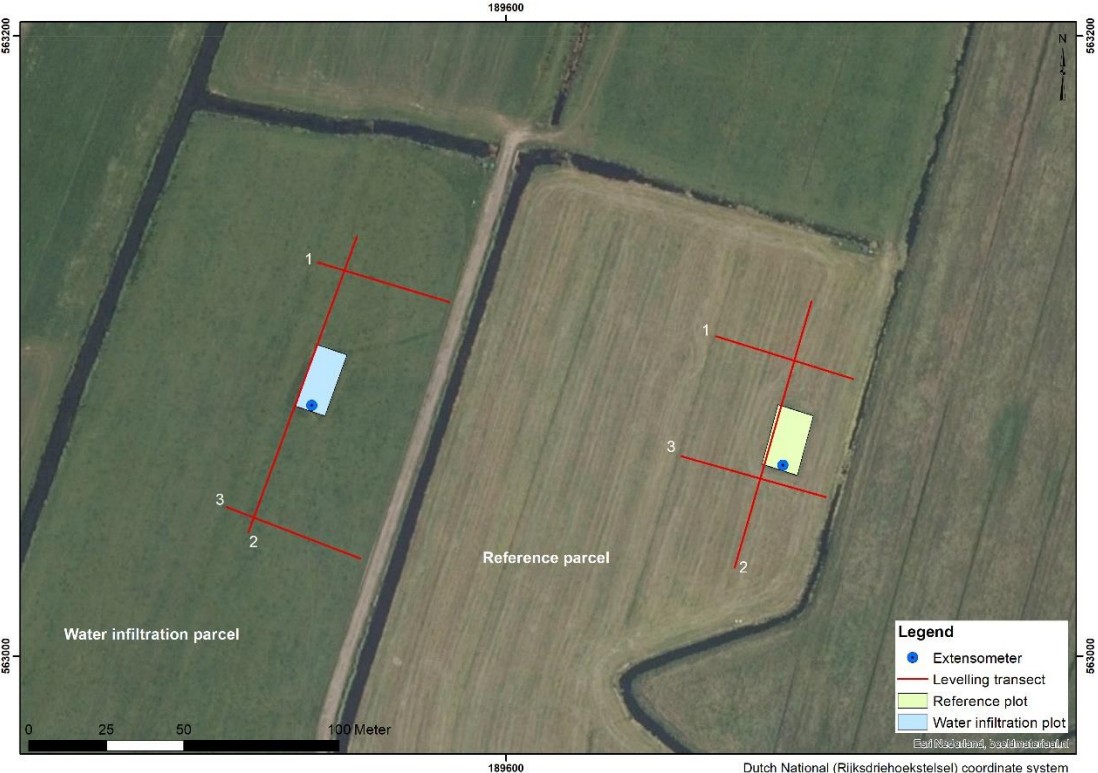

**Figure 4. Position of field plots, spirit levelling transects and extensometers in the reference and PWIS parcel at Aldeboarn.**

X, Y and Z coordinates and anchor depths of the extensometers in Aldeboarn are given in Table 1.






**Table 1. Positions and anchor depths of the extensometers in Aldeboarn. X and Y coordinates are given in the Dutch National Grid (Rijksdriehoekstelsel). Z (surface elevation) relative to Dutch Ordnance Datum (NAP). The anchor level positions are also indicated in the soil profile (black horizontal lines; green=clay, brown=peat, yellow=sand, orange=sand and/or loam).**

| Description | PWIS parcel | | Reference parcel | |
|---|---|---|---|---|
| X-coordinate [m] | 189537 | | 189689 | |
| Y-coordinate [m] | 563080 | | 563061 | |
| Surface elevation [m NAP] | -1.02 | | -1.02 | |
| Thickness top clay layer (m) | 0.4 | | 0.4 | |
| Thickness peat layers (m) | 1.6 | | 1.4 | |
| *Anchor depth [m below surface level]:* | | | | |
| Anchor 1 | -8.20 | | -7.40 | |
| Anchor 2 | -2.10 | | -2.10 | |
| Anchor 3 | -1.20 | | -1.21 | |
| Anchor 4 | -0.81 | | -0.80 | |
| Anchor 5 | -0.40 | | -0.40 | |
| Anchor 6 | -0.06 | | -0.06 | |


### 2.3.2 Rouveen

In Rouveen, drains are installed at circa 65 to 70 cm depth, in the longitudinal direction of the parcel. The drain spacing is ~8 m. The drains are connected to a submerged collector drain, which is directly connected to the ditch, making this a PWIS. Vertical soil movement has been measured by spirit levelling and extensometry since end 2018 (Figure 5). During the

measuring period, ditch water levels for both the reference and PWIS parcel have fluctuated roughly between 40 to 50 cm below surface level in summer and 50 to 60 cm below surface in winter (Van Asselen et al., 2023). Rouveen is an upward seepage location (see Van Asselen et al., 2023), output of the hydrological model LHM4.1 indicates average upward seepage for the period 2011-2018 of about 0.4 mm day$^{-1}$ at this location; LHM, 2023).





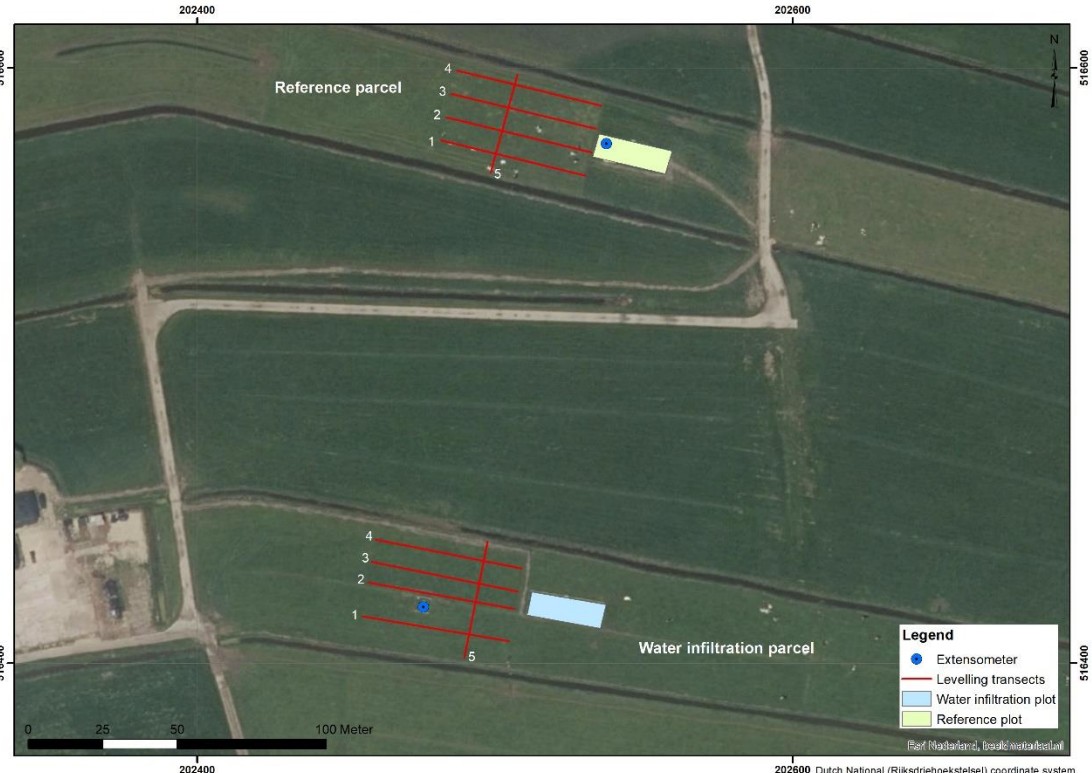

**Figure 5. Position of field plots, spirit levelling transects and extensometers in the reference and PWIS parcel at Rouveen.**

The subsurface in Rouveen is characterized by clay on peat on sand (see also soil profile figures at extensometer locations in Table 2 and Erkens et al., 2019). The top 0.05 to 0.10 m is composed of clayey peat with an organic matter content of ~30 %, below this layer a generally stiff marine clay is found until ~0.30 to ~0.40 m depth with an organic matter content of ~10 %. Below this clay layer peat occurs until a depth of 3.25 to 3.60 m below surface, the depth at which sandy Pleistocene deposits occur. The top of the peat layer is strongly amorphous. Within 0.80 m below surface a moss peat layer occurs of up to 0.20 m thick, containing sedge and wood remains. Below this layer a eutrophic sedge peat layer is found, that may contain reed and wood remains. At some locations within the parcels, the peat is a bit clayey at a depth of 2.20 to 2.45 m below surface.





**Table 2. Positions and anchor depths of the extensometers in Rouveen. X and Y coordinates are given in the Dutch National Grid (Rijksdriehoekstelsel). Z (surface elevation) relative to Dutch Ordnance Datum (NAP). The anchor level positions are also indicated in the soil profile (black horizontal lines; green=clay, brown=peat, yellow=sand, orange=sand and/or loam).**

| Description | PWIS parcel | | Reference parcel | |
|---|---|---|---|---|
| X-coordinate [m] | 202476 | | 202537 | |
| Y-coordinate [m] | 516419 | | 516574 | |
| Surface elevation [m NAP] | -0.56 | | -0.60 | |
| Thickness top clay layer (m) | 0.35 | | 0.35 | |
| Thickness peat layer (m) | 3.0 | | 3.0 | |
| *Anchor depth [m below surface level]:* | | | | |
| Anchor 1 | -4.2 | | -4.2 | |
| Anchor 2 | -1.15 | | -1.15 | |
| Anchor 3 | -0.35 | | -0.35 | |
| Anchor 4 | -0.05 | | -0.05 | |

### 2.3.3 Assendelft

In Assendelft an active WIS has been installed in 2017, aiming to keep the groundwater level at about 35 cm below surface level. Drains are installed at circa 50 to 60 cm depth, in the longitudinal direction of the parcel. The drain spacing is 4 m. Soil vertical movement has been measured using spirit levelling and extensometry since mid-2020 (Figure 6). Ditch water levels for both the reference and AWIS parcels are mostly 20 to 45 cm below surface level (Van Asselen et al., 2023).

The hydraulic head in Assendelft (water pressure in sandy Pleistocene deposits at 17 m depth) is higher than the phreatic groundwater levels in the reference parcel (Van Asselen et al., 2023). Upward seepage from this depth to the peat layer is however impeded by the thick (~10 m) marine (clay, loam and sand) deposits in between the peat and Pleistocene sand layers. Groundwater level measurements do indicate some upward seepage in dry periods at this location (Van Asselen et al., 2023). Also, the hydrological model LHM4.1 (LHM, 2023) indicates some upward seepage at this location (~0.1 mm day$^{-1}$ on average,

for the period 2011-2018).





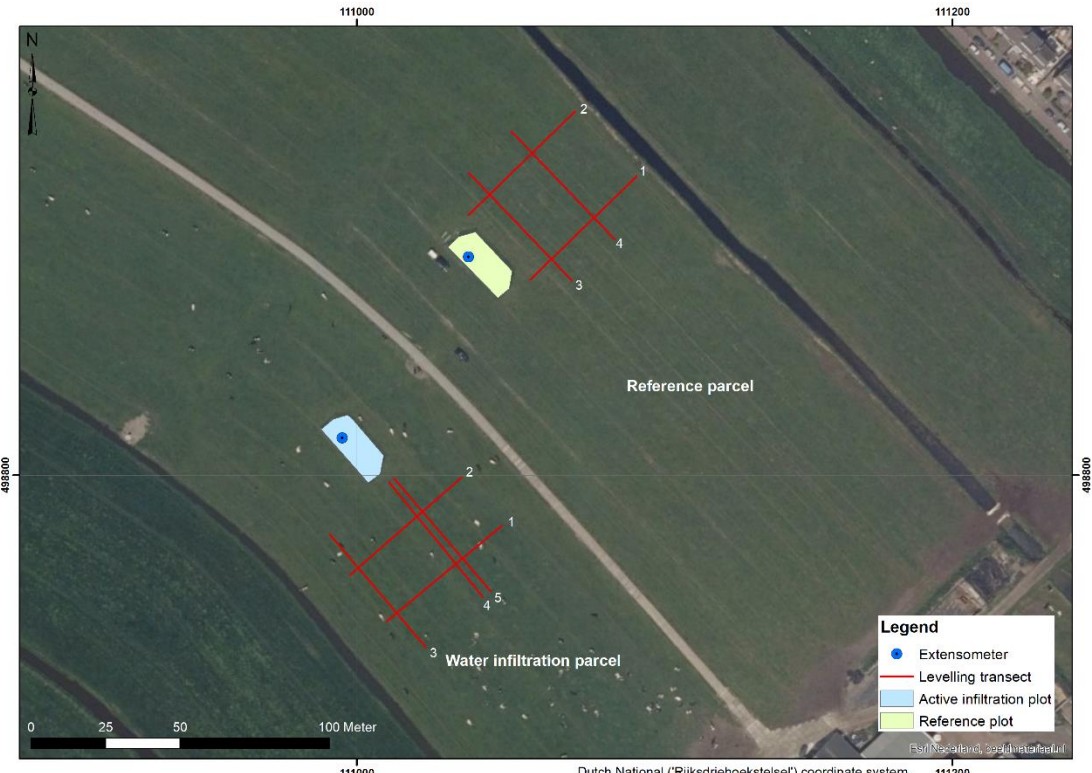

**Figure 6. Position of field plots, spirit levelling transects and extensometers in the reference and AWIS parcel at Assendelft.**

The Holocene sequence in Assendelft is about 16.5 m thick. In general, about 2 meters eutrophic reed-sedge peat is found on top of about ~14 m marine clayey and sandy deposits on ~0.5 m of basal peat (see also soil profiles at extensometer locations in Table 3 and Erkens et al., 2019). The organic matter content (LOI) in the top 20 to 30 cm roughly varies between 10 % and 40 %. This layer has consequently been classified as either an organic clay or a clayey peat. Below the basal peat layer, Pleistocene sandy deposits occur at a depth of 16.5 m below surface.

For X, Y and Z coordinates and anchor depths of the extensometers in Assendelft see Table 3.







**Table 3. Positions and anchor depths of the extensometers in Assendelft. X and Y coordinates are in the Dutch National Grid (Rijksdriehoekstelsel). Z (surface elevation) relative to Dutch Ordnance Datum (NAP). The anchor level positions are also indicated in the soil profile (black horizontal lines; green=clay, brown=peat, yellow=sand, orange=sand and loam).**

| Description | AWIS parcel | | Reference parcel | |
|---|---|---|---|---|
| X [m] | 110995 | | 111038 | |
| Y [m] | 498812 | | 498873 | |
| Surface elevation [m NAP] | -1.95 | | -1.88 | |
| Thickness top clayey peat layer (m) | 0.25 | | 0.30 | |
| Thickness peat layer (m) | 1.75 | | 1.75 | |
| *Anchor depth [m below surface level]:* | | | | |
| Anchor 1 | -16.80 | | -16.80 | |
| Anchor 2 | -1.99 | | -1.90 | |
| Anchor 3 | -1.22 | | -1.21 | |
| Anchor 4 | -0.81 | | -0.79 | |
| Anchor 5 | -0.34 | | -0.33 | |
| Anchor 6 | -0.06 | | -0.06 | |

### 2.3.4 Zegveld

In Zegveld multiple measures are investigated: both active and passive water infiltration systems in combination with either relatively high or low ditch water levels (Figure 7). Drains have been installed in 2016 at about 70-75 cm depth in the longitudinal direction of the parcels. The drain spacing is 6 m. At parcel 16, ditch water levels are relatively low; ~55 cm below surface level. The target groundwater level of AWIS 16 is 40 cm below surface level. At parcels 11 and 13 ditch water levels are kept relatively high at ~25 to 30 cm below surface level. The target groundwater level of AWIS 11 is 25 to 30 cm below surface level in summer and 35 to 40 cm below surface level in winter.



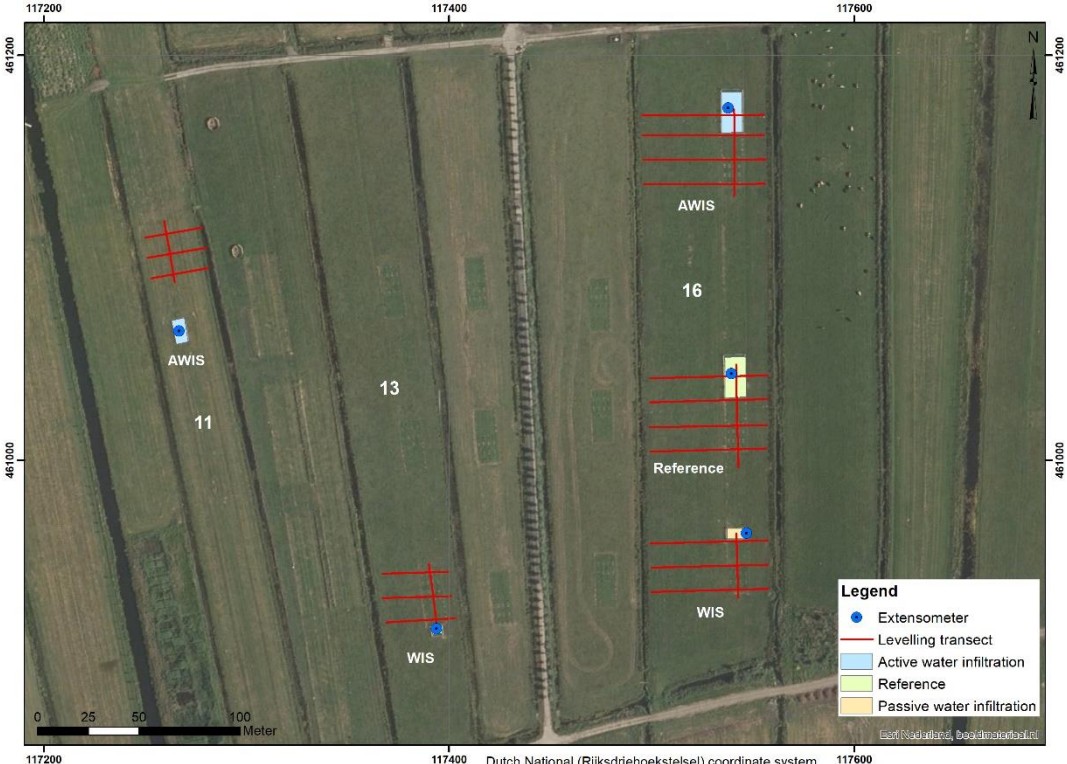

**Figure 7. Position of field plots, spirit levelling transects and extensometers in the reference and water infiltration parcels at Zegveld. In parcel number 16 ditch water levels are relatively low (around 55 cm below surface) and in parcel numbers 11 and 13 ditch water levels are relatively high (around 25 cm below surface).**

In Zegveld, a reliable automatic monitoring well for measuring the hydraulic head in the sandy Pleistocene subsurface is lacking. The hydrological LHM model indicates a near neutral situation at this location, with on average for the period 2011-2018 minor infiltration of -0,06 mm day$^{-1}$ (LHM, 2023).

The Holocene soft soil sequence in Zegveld is 6.10 to 6.35 m thick and consists predominantly of peat (see also soil profiles at extensometer location in Table 4 and Erkens et al., 2019). The top of underlying Pleistocene deposits consists of eolian cover sands. The top ~0.20 to ~0.50 m of the peat layer consists of clayey amorphous peat, with an organic matter content varying roughly between 30 and 40 %, further down gradually increasing to about 80 %. In parcel 16, a 10 to 15 cm thick anthropogenic organic layer may occur at the surface consisting of cow manure and dredge from the ditches, with variable sand admixture and/or pottery and brick remains (waste materials). To a depth of about 3 m predominantly wood peat occurs, below, eutrophic reed-sedge peat predominates, locally intercalated by thin reed peat layers. At a depth of 4.30 to 5.10 m below surface a few cm thick clayey interval occurs. Below this layer, the peat layer contains *Cladium mariscus* remains. At the transition to the Pleistocene sand wood peat may occur. Firm Pleistocene sand occurs at a depth of about 9 m below surface.





X, Y and Z coordinates and anchor depths of the extensometers in Zegveld are given in Table 4.

**Table 4. Positions and anchor depths of the extensometers in Zegveld. X and Y coordinates are given in the Dutch National Grid (Rijksdriehoekstelsel). Z (surface elevation) relative to Dutch Ordnance Datum (NAP). The anchor level positions are also indicated in the soil profile (black horizontal lines; green=clay, greenish brown = clayey peat or peaty clay, brown=peat, yellow=sand, 290 orange=sand and/or loam).**

| Description | AWIS parcel 16 | Reference parcel 16 | PWIS parcel 16 | PWIS parcel 13 | AWIS parcel 11 |
|---|---|---|---|---|---|
| X [m] | 117538 | 117539 | 117547 | 117394 | 117267 |
| Y [m] | 461174 | 461043 | 460964 | 460917 | 461064 |
| Surface elevation [m NAP] | -2.48 | -2.51 | -2.60 | -2.36 | -2.33 |
| Thickness top clayey peat layer (m) | 0.5 | 0.4 | 0.8 | 0.35 | 0.30 |
| Thickness peat layer* (m) | 5.8 | 6.0 | 5.35 | 5.95 | 6.15 |
| *Anchor depth [m below surface level]:* | | | | | |
| Anchor 1 | -9.20 | -9.30 | -9.20 | -9.95 | -9.98 |
| Anchor 2 | -4.21 | -4.49 | -4.26 | -4.72 | -6.48 |
| Anchor 3 | -1.21 | -1.20 | -1.18 | -4.31 | -5.05 |
| Anchor 4 | -0.80 | -0.79 | -0.78 | -0.82 | -0.82 |
| Anchor 5 | -0.42 | -0.41 | -0.40 | -0.41 | -0.38 |
| Anchor 6 | -0.07 | -0.06 | -0.04 | -0.05 | -0.05 |
| | soil composition | soil composition | soil composition | soil composition | soil composition |

*including intercalated thin clayey intervals at about 4.5 m depth.

### 2.3.5 Vlist

In Vlist drains have been installed in 2011 at circa 70 cm depth, perpendicular to the longitudinal direction of the parcel,
connected to one ditch. The drain spacing is 6 m. The drains are directly connected to the ditch, making this a passive WIS.
For this study, elevation (changes) has been measured by spirit levelling and extensometery since mid-2020 (Figure 8). The





ditch water level at both sides of the parcel has fluctuated between ~45 and ~60 cm below surface level (Van Asselen et al., 2023).

The relatively short data series of the hydraulic head available for Vlist (measured since October 2022) shows a decreasing trend if the phreatic groundwater level is relatively low and an increasing trend when phreatic groundwater levels are high (Van Asselen et al., 2023). This is an indication of some seasonal seepage/infiltration, although on average it is probably quite neutral, which is also indicated by the hydrological LHM model (~-0.03 mm day$^{-1}$ on average for the period 2011-2018; LHM, 2023).


The Holocene soft soil sequence in Vlist is 10 to 11 meters thick and consists of an alternation of (fluvial) clay and peat layers of in general several decimeters thick (see also soil profiles at extensometer locations in Table 5 and Erkens et al., 2019). Some of the clay layers contain sand. The top ~0.40 m consists of humic clay and/or strongly clayey peat, with an organic matter content ranging between roughly 15 and 35 % (highest at the top). Below the clayey top layer, a wood peat layer occurs, which 310 may be a bit clayey. At a depth of about 2 m below surface, a few dm thick clay layer is found, below which eutrophic (sedge, reed and or wood) peat layers alternate with clay layers.

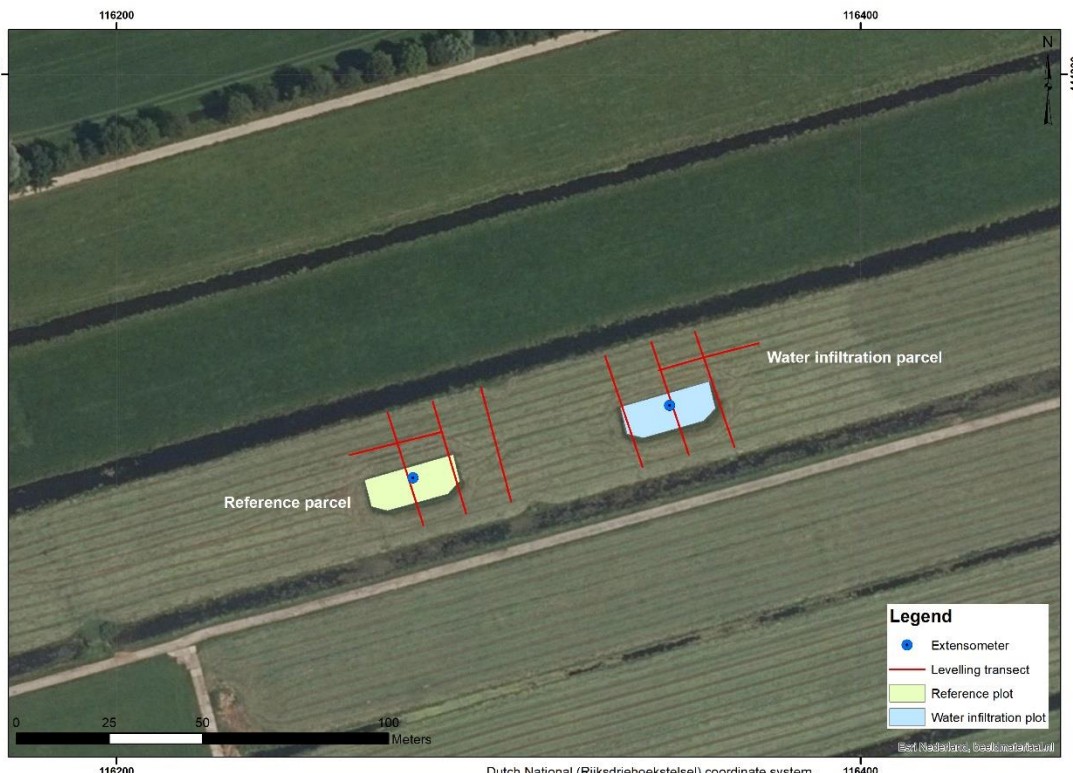

**Figure 8. Position of field plots, spirit levelling transects and extensometers in the reference and PWIS parcel at Vlist.**





X, Y and Z coordinates and anchor depths of the extensometers in Vlist are given in Table 5.

**Table 5. Positions and anchor depths of the extensometers in Vlist. X and Y coordinates are given in the Dutch National Grid (Rijksdriehoekstelsel). Z (surface elevation) relative to Dutch Ordnance Datum (NAP). The anchor level positions are also indicated in the soil profile (black horizontal lines; green=clay, brown=peat, yellow=sand, orange=sand and/or loam).**

| Description | PWIS parcel | | Reference parcel | |
|---|---|---|---|---|
| X [m] | 116349 | | 116280 | |
| Y [m] | 443911 | | 443891 | |
| Surface elevation [m NAP] | -1.70 | | -1.69 | |
| Thickness top clay layer (m) | 0.4 | | 0.4 | |
| Thickness first peat layer (m) | 1.6 | | 1.6 | |
| *Anchor depth [m below surface level]:* | | | | |
| Anchor 1 | -15.20 | | -15.30 | |
| Anchor 2 | -6.74 | | -6.85 | |
| Anchor 3 | -1.18 | | -1.20 | |
| Anchor 4 | -0.79 | | -0.78 | |
| Anchor 5 | -0.39 | | -0.37 | |
| Anchor 6 | -0.05 | | -0.05 | |


## 2.4 Data analysis

To get insight into seasonal vertical soil movement and long-term land subsidence, changes in elevation are calculated for each study site, for both spirit levelling and extensometer data, relative to the spirit levelling measurement day in January 2021 (time 0:00 for the hourly extensometer measurements). This reference was chosen because from this date onward, field
measurements are available for all five locations. If available, measurements before this date are included, also relative to the levelling measurement in January 2021.

*Spirit levelling data analysis*

For each spirit levelling field campaign (four times a year), measured surface elevations relative to the Dutch Ordnance Datum
('*Normaal Amsterdams Peil*'; NAP) are averaged for both the WIS and reference parcel. After calculating the elevation changes, the yearly (2021 and 2022) vertical movement dynamics (D) for year X have been derived from the maximum ($H_{max}$) and minimum ($H_{min}$) elevation (relative to January 2021) within one year, calculated for the period 1 November year X-1 until 31 October year X using:

$D = H_{max} - H_{min}$                                                                                                      (1)





In this way, the yearly vertical movement dynamics will in most cases represent seasonal subsidence going from the wet period during which the surface elevation is commonly highest (autumn and winter) to the dry period (spring and summer), when the surface elevation is commonly lowest.


*Extensometer data analysis*

In this study, extensometer anchors at approximately 0.05 and 0.80 m depth are included in the analysis. For Rouveen, an anchor at 0.80 m depth was not available, hence, the anchor at 1.15 m depth was used instead. Other anchor depths will be included in a future analysis. The (top) 0.05 m depth anchor is included because it best represents surface level, i.e., total

deformation of the soft soil sequence, and therefore best compares with spirit levelling measurements of the surface level. The ~0.80 m depth anchor was chosen to gain first insights into the relative contribution of the (predominantly) saturated soil below this level to vertical land surface movement. Elevation changes per anchor (hourly measurements) are calculated relative to the levelling field campaign day in January 2021 (at time 0:00). For each year, the relative contribution of the 0.80 or 1.15 m anchor to the 0.05 m (surface) anchor level was estimated based on the ratio of the dynamics of the 0.80 or 1.15 m anchor to

the dynamics of the surface anchor level (*100 %).

In addition, for each levelling field measurement day, the average elevation changes of the 0.05 and 0.80 m depth anchors have been calculated, also relative to the levelling field campaign day in January 2021. These daily extensometer averages are used to compare measurements from both techniques.


The land movement data are plotted together with local high-resolution phreatic groundwater level measurements (for details see Van Asselen et al., 2023). At WIS parcels, vertical land movement is compared with the phreatic groundwater dynamics from a monitoring well halfway between two drains (similar position as the extensometer). For the reference parcel, the phreatic groundwater level monitoring well closest to the extensometer has been used.

**3 Soil vertical movement measurement results**

**3.1 Aldeboarn**

Both levelling and extensometer measurements demonstrate less vertical dynamics in the PWIS parcel compared to the reference parcel (3 to 20 mm less seasonal subsidence; Table 6; Figure 9). The summer levelling measurement in 2021 is lacking but the extensometer data series shows relatively low summer subsidence for this period (21 mm; Table 6; Fig. 9),

presumably related to the relatively wet summer of 2021 with relatively high phreatic groundwater levels. In the dry and warm summer of 2022 the phreatic groundwater level lowered much more. Likewise, vertical dynamics (i.e. seasonal subsidence) were higher in 2022 than in 2021. The highest vertical dynamics has been measured in the reference parcel in 2022 (58 mm),



based on the extensometer data series. Based on the extensometer data, we see a higher absolute effect on vertical dynamics
of the PWIS in a dry year (2022) compared to a wet year (2021). Furthermore, the extensometer data indicates that the
contribution of the -0.80 m anchor to seasonal subsidence of the -0.06 m anchor is higher in the reference parcel than in the
PWIS parcel.

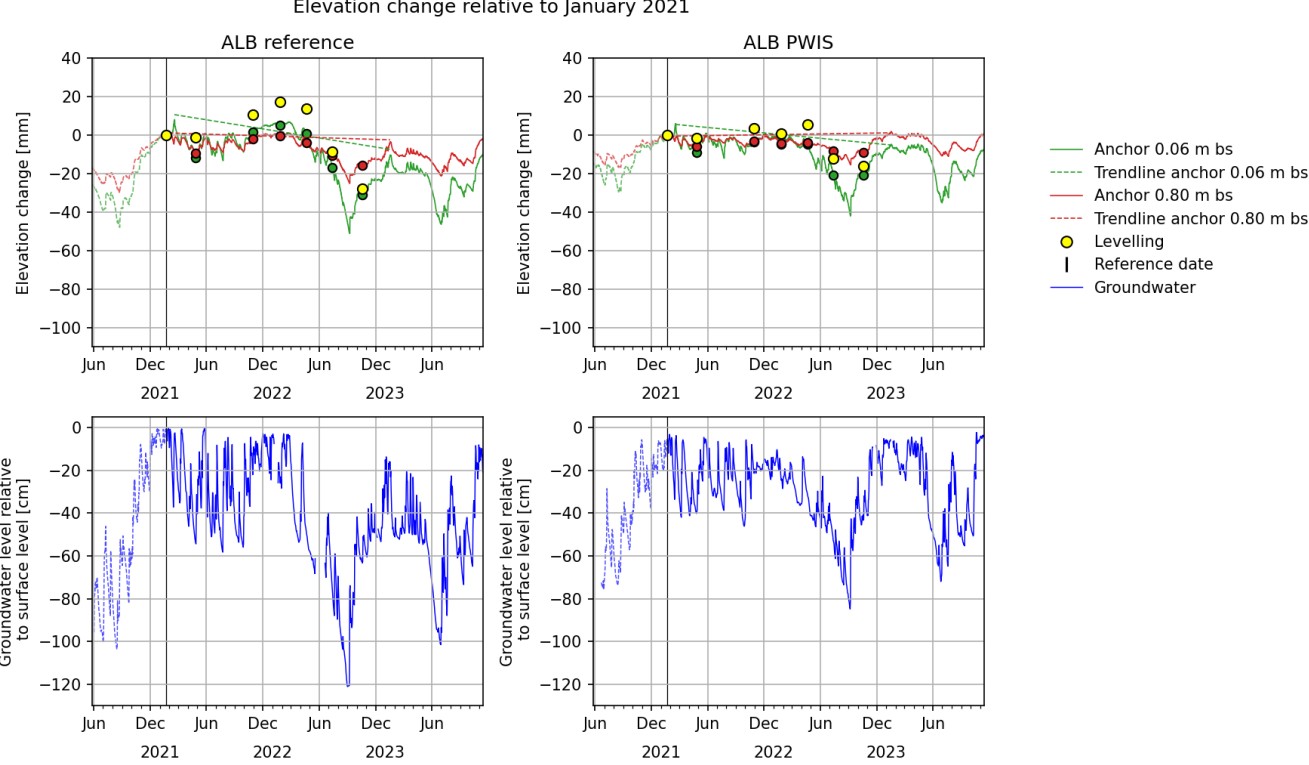

**Figure 9. Elevation changes relative to the levelling day in January 2021, indicated with the black vertical line, for the reference
(ALB RF; left) and WIS (ALB WIS; right) parcel in Aldeboarn, based on spirit levelling (average of all measurements points per
parcel) and extensometer measurements (surface anchor at 0.06 m depth in green and anchor at ~0.80 m depth in red; daily averages
for the levelling days are indicated with green and red dots respectively). In blue the phreatic groundwater level is shown. Bs = below
surface. Data before the reference date are indicated with dotted lines. Standard deviation for average elevation changes per parcel
based on levelling is in the order of 10 to 20 mm.**







**Table 6. Yearly vertical soil dynamics in mm for Aldeboarn, based on levelling and extensometer measurements (anchors at 0.06 and 0.80 m depth). For levelling, the summer measurement is lacking for 2021, therefore the spring measurement (end of April) has been used instead for calculating D. RF=reference parcel, Dif.=difference. Extensometer daily averages have been calculated for the days of the levelling measurements. The contribution of the 0.80 m anchor to the surface anchor (0.06 m depth) is given in the last column.**

| Yearly vertical soil dynamics (mm): | | Levelling | Extensometer; Anchor depth 0.06 m | | Extensometer; Anchor depth 0.80 m | | Contribution -0.80 anchor to -0.06 anchor |
|---|---|---|---|---|---|---|---|
| | | | *Average on levelling day* | *Full data series* | *Average on levelling day* | *Full data series* | |
| **2021** | RF | 5 | 14 | 21 | 9 | 10 | 48 % |
| | WIS | 1 | 9 | 18 | 6 | 7 | 36 % |
| | *Dif. to RF* | *-4 (=- 83%)* | *-5 (=- 34%)* | *-3 (=- 15%)* | *-4 (=- 39%)* | *-4 (=-36 %)* | |
| **2022** | RF | 46 | 36 | 58 | 15 | 26 | 45 % |
| | WIS | 28 | 16 | 42 | 5 | 15 | 36 % |
| | *Dif. to RF* | *-18 (=- 38%)* | *-20 (=-55 %)* | *-16 (=-28 %)* | *-10 (=-67 %)* | *-11 (=-42 %)* | |

### 3.2 Rouveen

In general, the PWIS in Rouveen has resulted in lower phreatic groundwater levels compared to the reference parcel (Van Asselen et al., 2023; Figure 10). This undesired effect of the PWIS was attributed to substantial upward seepage in this area. Commonly, upward seepage causes relatively high phreatic groundwater levels, also in the summer season. The drains, that were installed at a depth close to the deepest annual phreatic groundwater level, have predominantly drained groundwater. Their role as infiltration drains has been minimal at this location. Field measurements do not show a clear effect of the PWIS on soil movement dynamics; in general, differences in dynamics between the reference and PWIS are relatively small and on several occasions in opposite direction from what was expected (Table 7). The highest amount of vertical dynamics recorded at this location is 53 mm, measured by spirit levelling in the WIS parcel in 2021.

In both parcels in Rouveen, the four years of extensometer measurements may be used for a first rough estimate of long-term (multiyear) soil subsidence. A longer time series is however needed to be able to better filter out effects of yearly variations in dynamics caused by yearly variations in precipitation and evapotranspiration, and hence, to make more reliable estimates of land subsidence. Still, a linear trendline was fitted on the yearly highest (winter) elevations (January and February) of the successive years of the extensometer surface (-0.05 m depth) anchor, resulting in a trendline with a slope of -4 mm yr$^{-1}$ ($R^2$=0.91) subsidence for the reference parcel and -13 mm/yr ($R^2$=0.95) for the PWIS parcel (Figure 10). The same was done for the 1.15 m anchor, resulting in a slope of 0 mm/yr in the reference parcel ($R^2$=0.07) and -6 mm yr$^{-1}$ ($R^2$=0.86) in the PWIS parcel. Because the time series is relatively short, derived subsidence rates should be regarded with caution. Still, the data so





far suggest more subsidence in the PWIS parcel as compared to the reference parcel, which could be explained by the lower phreatic groundwater levels measured in this parcel, causing increased peat decomposition and compaction.

The need for longer timeseries is also indicated by the levelling data, which so far do not show a clear long-term trend at all.
Currently, the linear trendline fitted through the five winter measurements of the levelling data timeseries is still strongly influenced by one – possibly deviating – measurement, and the variation is high (low $R^2$; reference parcel: slope = 4 mm yr$^{-1}$, $R^2 = 0.27$; PWIS parcels: slope = -2 mm yr$^{-1}$, $R^2 = 0.06$; not presented in graphs; see also paragraph 4.2).

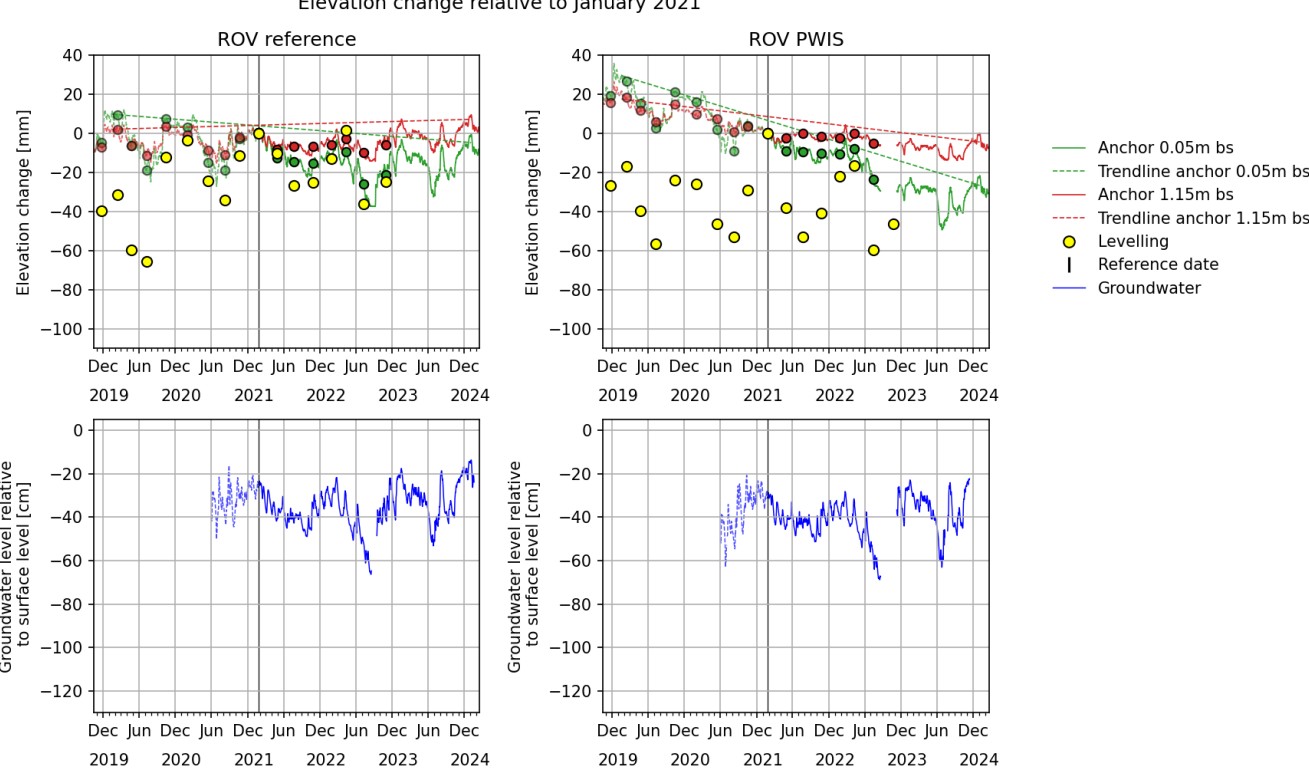

**Figure 10. Elevation changes relative to the levelling day in January 2021, indicated with the black vertical line, for the reference (ROV RF; left) and WIS (ROV WIS; right) parcels in Rouveen, based on spirit levelling and extensometer measurements (surface anchor at 0.05 m depth in green and anchor at 1.15 m depth in red; daily averages for the levelling days are indicated with green and red dots respectively). In blue the phreatic groundwater level is shown. Data before the reference date are indicated with dotted lines or lighter colors. Standard deviation for elevation changes based on levelling is in the order of 10 to 20 mm. Trendlines are**
**fitted based on the highest elevations of the extensometer anchors in the succeeding winters (January and February). Slopes of the 0.05 anchor trendlines are -4 and -13 mm yr$^{-1}$, with $R^2$ of 0.91 and 0.95, for the reference and WIS parcel respectively. Slopes of the 1.15 anchor trendlines are 0 and -6 mm yr$^{-1}$, with $R^2$ of 0.07 and 0.86, for the reference and WIS parcel respectively. bs=below surface.**






**Table 7. Yearly vertical dynamics in mm for Rouveen, based on levelling and extensometer measurements. RF=reference parcel, Dif.=difference. Extensometer daily average has been calculated for the day of the levelling measurement. The contribution of the 1.15 m anchor to the surface anchor (0.05 m depth) is given in the last column.**


| Vertical soil dynamics (mm): | | Levelling | Extensometer; Anchor depth 0.05 m | | Extensometer; Anchor depth 1.15 m | | Contribution of -1.15 anchor to -0.05 anchor |
|---|---|---|---|---|---|---|---|
| | | | *Average on levelling day* | *Full data series* | *Average on levelling day* | *Full data series* | |
| **2019** | RF | 34 | 28 | 37 | 15 | 20 | 53 % |
| | WIS | 40 | 24 | 35 | 12 | 23 | 65 % |
| | *Dif. to RF* | *+5 (+16 %)* | *-4 (-16 %)* | *-2 (-5 %)* | *-3 (-19 %)* | *+3 (+15 %)* | |
| **2020** | RF | 31 | 23 | 37 | 11 | 20 | 54 % |
| | WIS | 27 | 25 | 32 | 9 | 15 | 45 % |
| | *Dif. to RF* | *-3 (-11 %)* | *+3 (+12 %)* | *-5 (-13 %)* | *-2 (-15 %)* | *-6 (-28 %)* | |
| **2021** | RF | 26 | 15 | 26 | 8 | 15 | 57 % |
| | WIS | 53 | 10 | 20 | 3 | 9 | 45 % |
| | *Dif. to RF* | *+27 (+101 %)* | *-5 (-32 %)* | *-6 (-23 %)* | *-5 (-68 %)* | *-6 (-39 %)* | |
| **2022** | RF | 38 | 17 | 34 | 7 | 17 | 48 % |
| | WIS | 44 | 16 | 27 | 5 | 11 | 40 % |
| | *Dif. to RF* | *+6 (+15 %)* | *-1 (-4 %)* | *-7 (-21 %)* | *-2 (-26 %)* | *-6 (-35 %)* | |

## 3.3 Assendelft

The AWIS in Assendelft has caused higher summer phreatic groundwater levels (Figure 11; see also Van Asselen et al., 2023) and less vertical soil dynamics compared to the reference parcel. The effect on vertical dynamics (i.e., seasonal subsidence) is especially notable in the dry summer of 2022 (Figure 11; Table 8). The highest vertical dynamics recorded is 79 mm, based

on the extensometer data series (-0.06 m anchor) in the reference parcel. The contribution of the -0.80 m anchor is relatively high (>50 %), especially in the wet year of 2021.

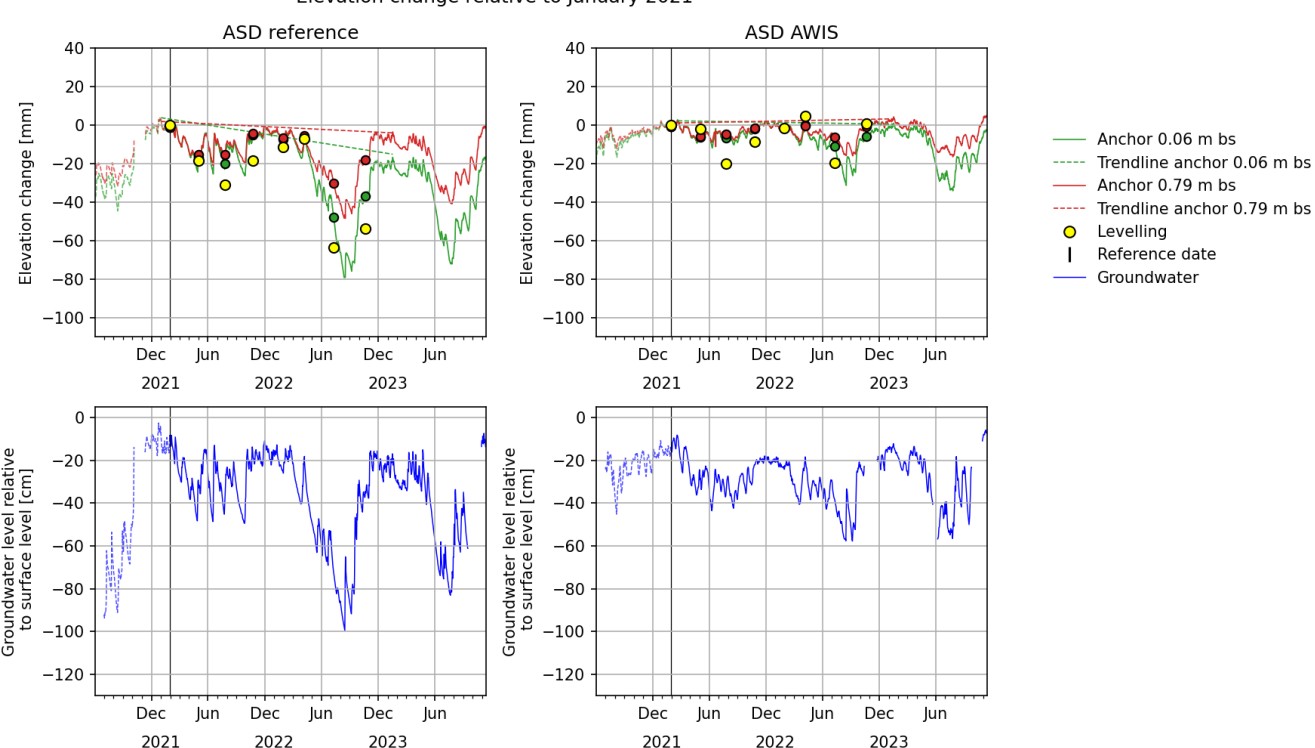

**Figure 11. Elevation changes relative to the levelling day in January 2021, indicated by the black vertical line, for the reference (ASD RF; left) and AWIS (ASD AWIS; right) parcels in Assendelft, based on spirit levelling and extensometer measurements (surface anchor at 0.06 m depth in green and anchor at ~0.80 m depth in red; daily averages for the levelling days are indicated with green and red dots respectively). In blue the phreatic groundwater level is shown. Data before the reference date are indicated with dotted lines. Standard deviation for elevation changes based on levelling is in the order of 10 to 20 mm. bs=below surface.**

**Table 8. Yearly vertical soil dynaimcs in mm for Assendelft, based on levelling and extensometer measurements. RF=reference parcel, Dif.=difference. Extensometer daily averages have been calculated for the days of the levelling measurements. The contribution of the 0.80 m anchor to the surface anchor (0.06 m depth) is given in the last column.**

| Vertical soil dynamics (mm): | | Levelling | Extensometer; anchor depth 0.06 m | | Extensometer; Anchor depth 0.80 m | | Contribution -0.80 anchor to -0.06 anchor |
|---|---|---|---|---|---|---|---|
| | | | *Average on levelling day* | *Full data series* | *Average on levelling day* | *Full data series* | |
| 2021 | RF | 31 | 19 | 30 | 15 | 25 | 85 % |
| | AWIS | 20 | 6 | 15 | 6 | 13 | 92 % |
| | *Dif. to RF* | *-11 (=-36 %)* | *-13 (=-67 %)* | *-15 (=-51 %)* | *-10 (=-63 %)* | *-12 (=-74 %)* | |
| 2022 | RF | 57 | 42 | 79 | 25 | 48 | 61 % |
| | AWIS | 24 | 11 | 33 | 6 | 18 | 54 % |
| | *Dif. to RF* | *-32 (=-57 %)* | *-31 (=-74 %)* | *-45 (=-58 %)* | *-19 (=-77 %)* | *-31 (=-6 3%)* | |





### 3.4 Zegveld

In Zegveld, measurements took place in five different parcels. In general, the different measures, i.e., type of WIS and high or low ditch water level, have resulted in higher summer phreatic groundwater levels and less vertical soil dynamics compared

to the reference parcel, as demonstrated by all types of elevation measurements applied at this study site (Table 9; Figure 12; Fig. S1; Table S1). The contribution of the -0.80 cm anchor is relatively high at Zegveld (> ~60 %) and is highest in the reference parcel (86 %).

Yearly vertical soil dynamics is highest in the reference parcel. In the summer of 2022, the extensometer full data series

recorded total dynamics of 98 mm in this parcel relative to the preceding winter. For most cases, vertical dynamics was higher in the dry year of 2022 than in the wetter year of 2021. No clear relation between the type of measure and the calculated (reductions in) dynamics have been observed, but they did all show different vertical dynamics as compared to the reference parcel. Manual phreatic groundwater level measurements in 2019 and 2020 did show that summer phreatic groundwater levels were highest in the AWIS parcel, followed by the PWIS parcel, and were lowest in the reference parcel (Erkens et al., 2021).

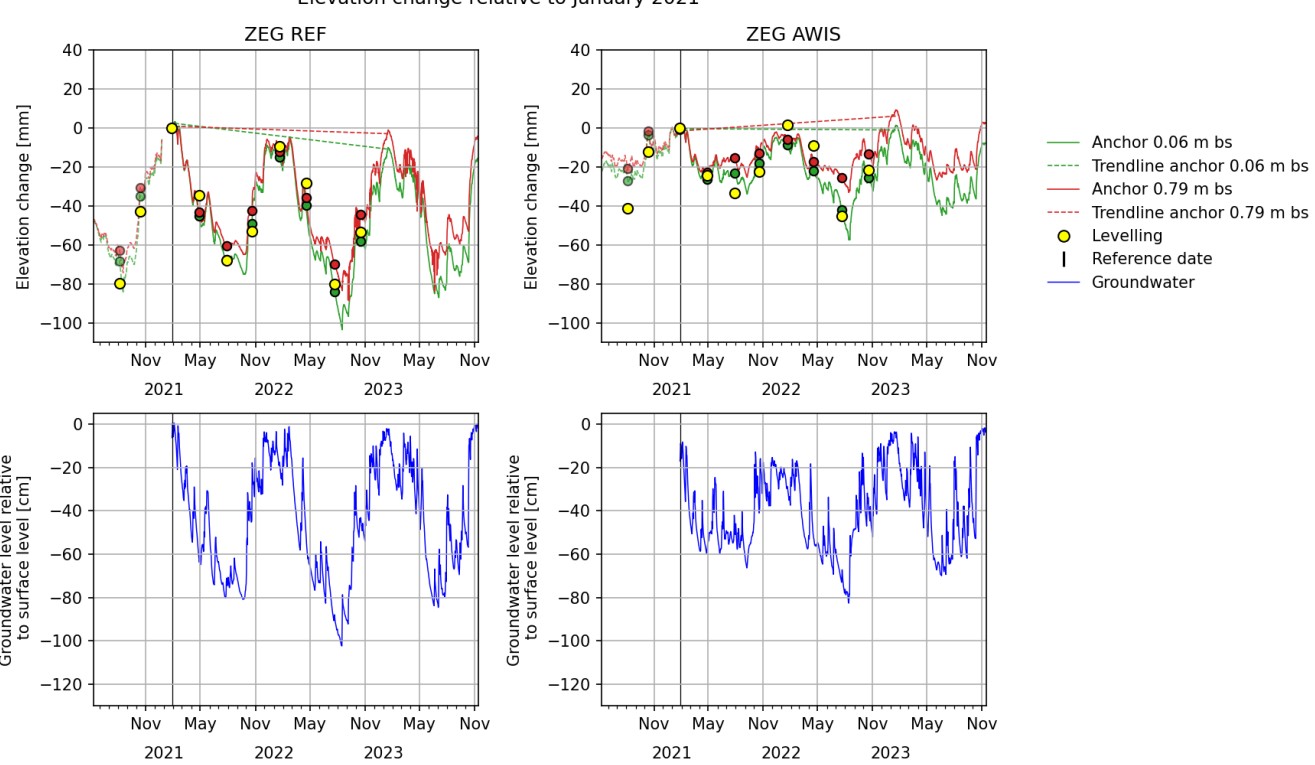


**Figure 12. Elevation changes relative to the levelling day in January 2021, indicated by the black vertical line, for two parcels in Zegveld (in parcel 16 with low ditch water level: ZEG REF = reference parcel, and ZEG AWIS), based on spirit levelling and extensometer measurements (surface anchor at ~0.05 m depth in green and anchor at ~0.80 m depth in red; daily averages for the levelling days are indicated with green and red dots respectively). In blue the phreatic groundwater level is shown. Data before the**





reference date are indicated with dotted lines. Standard deviation for elevation changes based on levelling is in the order of 10 to 20 mm. bs= below surface.

**Table 9. Yearly vertical soil dynamics in mm for Zegveld, based on levelling and extensometer measurements. RF=reference parcel, Dif.=Difference. Extensometer daily averages have been calculated for the days of the levelling measurements. The contribution of the 0.80 m anchor to the surface anchor (0.05 m depth) is given in the last column.**

| Vertical soil dynamics (mm): | | Levelling | Extensometer; Anchor depth 0.05 m | | Extensometer; Anchor depth 0.80 m | | Contribution -0.80 anchor to -0.05 anchor |
|---|---|---|---|---|---|---|---|
| | | | *Average on levelling day* | *Full data series* | *Average on levelling day* | *Full data series* | |
| **2021** | p16-RF | 68 | 66 | 78 | 59 | 68 | 86 % |
| | p16-AWIS | 33 | 26 | 36 | 22 | 27 | 75 % |
| | *Dif. to RF* | *-35 (-51 %)* | *-40 (-61 %)* | *-43 (-54 %)* | *-37 (-63 %)* | *-41 (-61 %)* | |
| **2022** | p16-RF | 71 | 69 | 98 | 57 | 84 | 86 % |
| | p16-AWIS | 47 | 33 | 53 | 19 | 31 | 59 % |
| | *Dif. to RF* | *-24 (-34 %)* | *-36 (-52 %)* | *-45 (-46 %)* | *-38 (-67 %)* | *-53 (-63 %)* | |

**3.5 Vlist**

In general, measurements from Vlist indicate slightly higher summer phreatic groundwater levels (Van Asselen et al., 2023) and slightly lower vertical dynamics in the PWIS parcel compared to the reference parcel (up to 10 mm less vertical dynamics; Table 10; Figure 13). Most pronounced effects are measured based on surface levelling. The highest amount of vertical dynamics at this location is 47 mm, which has been measured by spirit levelling in the reference parcel in 2022. The

contribution of the -0.80 m anchor to the surface anchor is lower in the dry year of 2022.





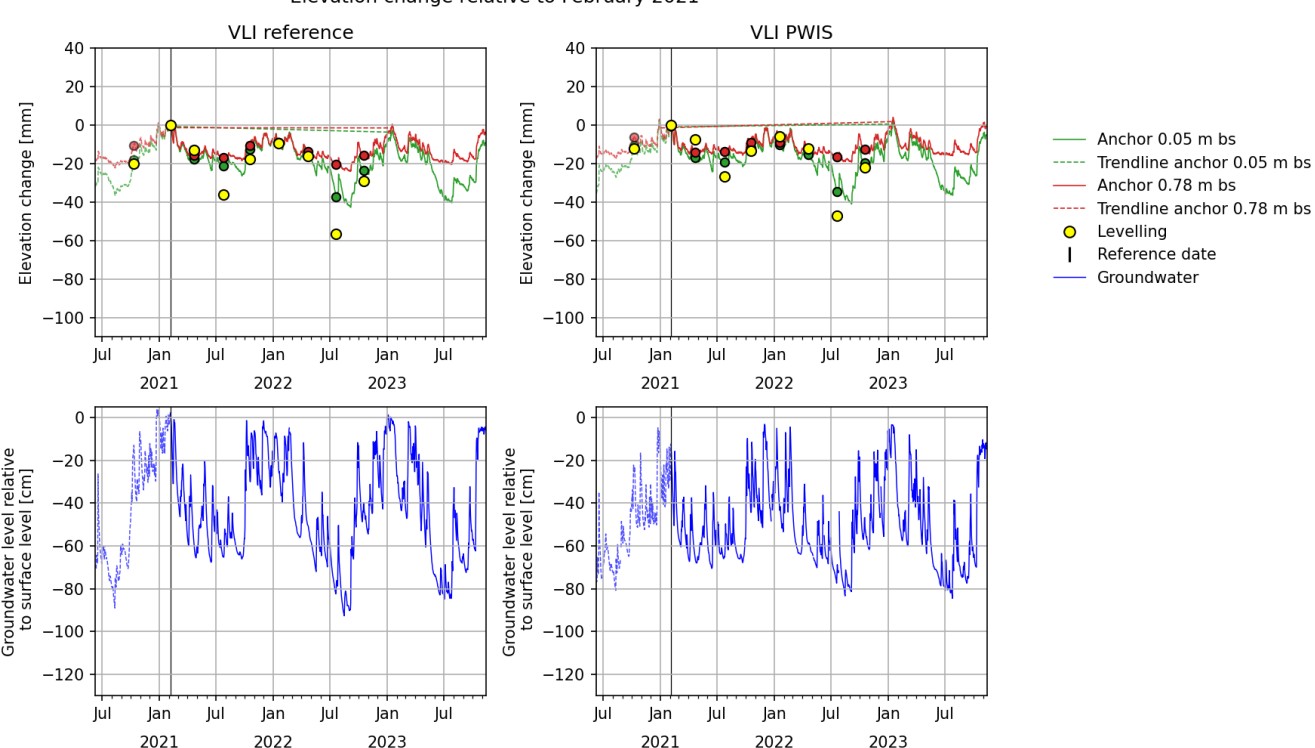

**Figure 13. Elevation changes relative to the levelling day in January 2021, indicated by the black vertical line, for the reference (VLI RF; left) and PWIS (VLI WIS; right) parcels in Vlist, based on spirit levelling and extensometer measurements (surface anchor at 0.05 m depth in green and anchor at ~0.80 m depth in red; daily averages for the levelling days are indicated with green and red dots respectively). In blue the phreatic groundwater level is shown. Data before the reference date are indicated with dotted lines. Standard deviation for elevation changes based on levelling is in the order of 10 to 20 mm. bs=below surface.**

**Table 10. Yearly vertical soil dynamics in mm for Vlist, based on levelling and extensometer measurements. RF=reference parcel, Dif.=difference. Extensometer daily averages have been calculated for the days of the levelling measurements. Standard deviation for elevation changes based on levelling is on the order of 10 to 20 mm. The contribution of the 0.80 m anchor to the surface anchor (0.05 m depth) is given in the last column.**

| Vertical soil dynamics (mm): | | Levelling | Extensometer; Anchor depth 0.05 m | | Extensometer; Anchor depth 0.80 m | | Contribution -0.80 anchor to -0.05 anchor |
|---|---|---|---|---|---|---|---|
| | | | *Average on levelling day* | *Full data series* | *Average on levelling day* | *Full data series* | |
| 2021 | RF | 36 | 22 | 28 | 17 | 20 | 70 % |
| | WIS | 27 | 19 | 27 | 15 | 19 | 69 % |
| | *Dif. to RF* | *-10 (-27 %)* | *-2 (-10 %)* | *-1 (-4 %)* | *-2 (-13 %)* | *-1 (-5 %)* | |
| 2022 | RF | 47 | 28 | 39 | 11 | 20 | 51 % |
| | WIS | 42 | 24 | 40 | 7 | 18 | 45 % |
| | *Dif. to RF* | *-6 (-12 %)* | *-4 (-13 %)* | *+1 (+2 %)* | *-4 (-33 %)* | *-2 (-10 %)* | |



### 3.6 Main findings of vertical land movement measurements

- At all locations except for Rouveen, the WIS has resulted in less lowering of the phreatic groundwater level in summer and in reduced yearly vertical dynamics, i.e., seasonal subsidence. Compared to the reference parcels, the reduction of the yearly vertical soil dynamics is up to 77 % (measured at Assendelft in the summer of 2022 at the anchor level of ~80 cm depth, Assendelft is also the location where highest effects on summer phreatic groundwater levels were measured (Van Asselen et al., 2023); see also Table 6, Table 7-Table 9). In absolute numbers, the AWIS in Assendelft resulted in up to 45 mm less seasonal subsidence compared to the reference parcel.

- The highest yearly vertical soil dynamics, i.e., seasonal subsidence, has been recorded by the extensometer full data series in the summer of 2022 in the reference parcel of Zegveld (98 mm).

- Deformation of the saturated subsurface (i.e., soil below ~0.80 m depth, or in case of Rouveen below ~1.15 m depth) contributes considerably to surface level vertical movement. The contribution of deformation of the soil below this anchor level to surface subsidence is about 30 to 95 % (based on extensometer measurements, Table 6, Table 7-Table 9).

- The yearly vertical dynamics derived from the daily averaged extensometer data are usually lower than the dynamics calculated from corresponding spirit levelling data (Table 6, Table 7-Table 9). This may be caused by deformation of the top ~5 cm, which is the depth of the top anchor of the extensometer, and hence, this deformation is not measured by the extensometer. With spirit levelling, the surface level is measured.

- The full extensometer data series, however, often yield higher vertical dynamics than those inferred from the spirit levelling (e.g., Aldeboarn, Assendelft, Zegveld). This suggests that quarterly measurements often miss most extreme elevations (minima and maxima) occurring within a year, which is indeed clearly noticeable when comparing the levelling data points (yellow) and the extensometer full data series in Figs. Figure 9 andFigure 13.

- Differences between levelling and extensometer surface level measurements may also be caused by spatial variations in surface level movement: the levelling data are spatial averages, while the extensometer measures at one location.

- Only for Rouveen, for which the longest time series was available, the multiyear subsidence rate was estimated (4 mm yr$^{-1}$ in the reference parcel, 13 mm yr$^{-1}$ in the PWIS parcel; note the higher subsidence rate estimated for the PWIS parcel at this upward seepage area; the PWIS resulted in lower summer groundwater levels due to increased drainage of seepage water). At the other locations, longer data series are needed to estimate subsidence rates.





## 4. Evaluation and discussion

### 4.1 Effects of water infiltration systems on vertical soil movement

### 4.1.1 The relation between the phreatic groundwater level and soil deformation

At all locations except for Rouveen, reduced vertical soil dynamics have been observed in the WIS parcel, compared to the adjoining reference parcel. The key factor causing this is presumably the influence of the WIS on the phreatic groundwater level. We observe a clear relation between the phreatic groundwater level and vertical soil dynaimcs. For the five investigated

study locations, except for Rouveen, the WIS have reduced phreatic groundwater level lowering in summer periods (Figure 9 to 13; Van Asselen et al., 2023), which commonly coincides with less seasonal subsidence in summer.

Figure 14 shows the relation between the yearly deepest phreatic groundwater level and vertical dynamics of the top extensometer anchor. For most parcels, except for those in Rouveen, a positive relation between the deepest groundwater level

and vertical dynamics, i.e. seasonal subsidence, is observed. Furthermore, results show that in the summer of 2022, phreatic groundwater levels dropped much deeper than in 2021, which coincides with generally more seasonal subsidence in 2022 compared to 2021. Results of this study also demonstrate that phreatic groundwater level fluctuations at shorter (daily/weekly) timescales also affect vertical soil movement; short-term peaks of the phreatic groundwater level often correspond with short-term peaks of vertical soil movement (Figure 9 to 13).





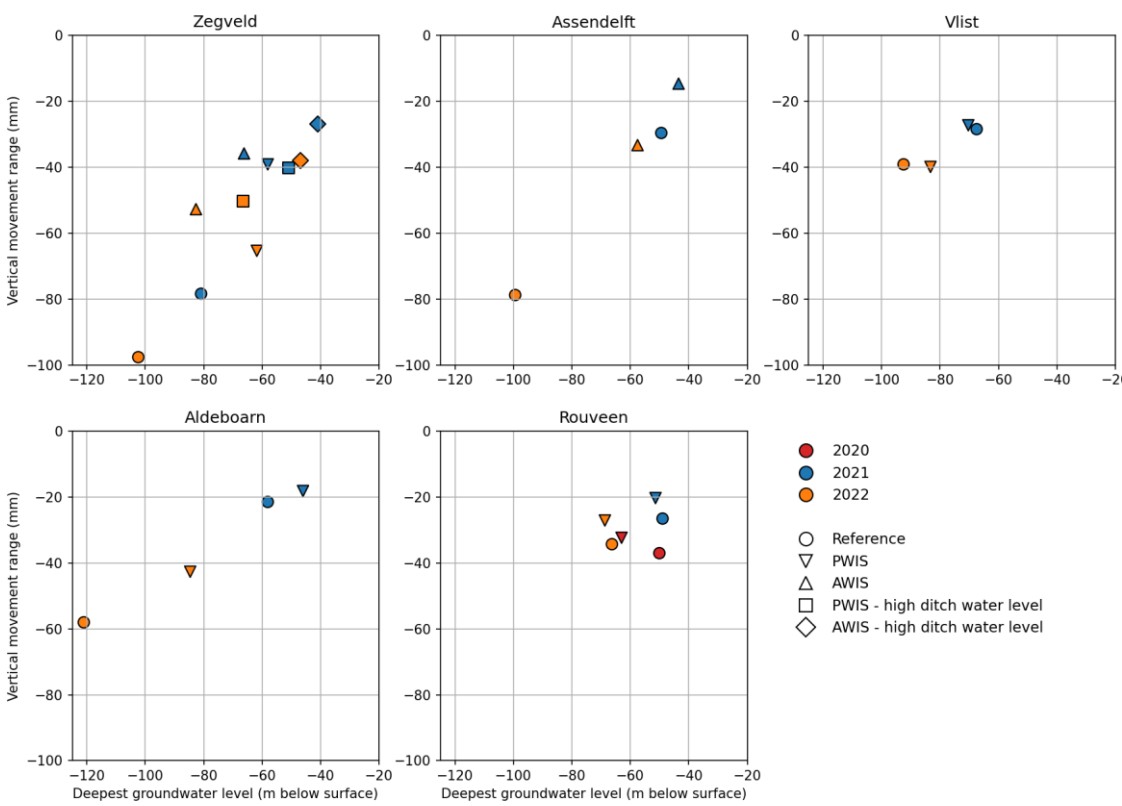

**Figure 14. Relation between the yearly vertical dynamics and deepest groundwater level for all parcels at all five locations.**

In Rouveen, which is located in an area with upward seepage, the drains have unintendedly increased drainage, resulting in lower phreatic groundwater levels as compared to the reference situation. Extensometer measurements indicate that this has resulted in more long-term subsidence in the PWIS parcel as compared to the reference parcel, which also demonstrates the significant influence of phreatic groundwater level dynamics on vertical land movement dynamics. Results for Rouveen also indicate that for estimating a long-term subsidence rate based on levelling measurements a longer time series is needed (section 3.2). The levelling measurements record the elevation for one particular moment, for which the surface may be relatively high (if wet conditions prevail) or may be relatively low (in case of relatively dry conditions), and the measurement moment do not necessarily correspond to the actual highest and lowest elevations. For the currently availably time series, with five winter measurements that were used for fitting the linear trend, this still causes an uncertain trend and a low $R^2$ (high variation). A single measurement of the time series clearly still has too much influence on the fitted trend (section 3.2). Longer levelling timeseries (preferably at least ten years) will result in better fits (e.g., Massop et al., to be submitted).





### 4.1.2. Suppression of both unsaturated zone shrink-swell and saturated poroelastic deformation

The effect of WIS on the phreatic groundwater levels affect short-term vertical movements (i.e., reversible deformation) of the
soil both above and below ~0.80 m depth (Table 6, Table 7-Table 9). Above this depth the soil is commonly unsaturated and
reversible deformation is mostly caused by shrink and swell processes. Less deep lowering of the phreatic groundwater level
in summer limits shrinkage of the soil because the water content remains higher and suction forces remain lower compared to
soils with lower summer groundwater levels. Part of the short-term vertical downward movement caused by shrinkage is
reversible; the soil swells again when conditions get wetter (Figure 9 to 13).


Less deformation of the soil in the commonly saturated zone below ~0.80 m depth may be explained by effects on the effective
stress in the subsurface. The effective stress, changes of which causes deformation of soil, is the total stress (i.e., weight) minus
the pore water pressure (Terzaghi, 1943). In a situation in which a WIS causes less deep lowering of the phreatic groundwater
level, the pore water pressure is reduced less compared to a situation with a deeper groundwater level. A smaller reduction in
pore water pressure results in a smaller increase in the effective stress (the total stress will not change much as a consequence
of groundwater level fluctuations), resulting in less deformation. Reversible deformation in the saturated soil is referred to as
poroelastic deformation (e.g., Kümpel, 1991). In Zegveld, the contribution of poroelastic deformation to surface level vertical
movement is relatively large, presumably due to the relatively thick (~6 m) peat layer that is prone to deformation at this
location. The total thickness of soil layers in which poroelastic deformation occurs often is much larger than the thickness of
the unsaturated zone.

The role of the depth of the phreatic groundwater level is supported by this study, showing that in the relatively dry summer
of 2022 there was more groundwater level lowering and seasonal subsidence, as compared to the relatively wet summer of
2021.

**4.1.3 Suppression of subsidence**

In situations where WIS cause less deep summer phreatic groundwater levels for shorter periods in peat meadows, it is expected
that pore water pressures remain higher, and the effective stress remains low (if loading conditions do not change). Hence, the
risk for irreversible compaction will be lower. In addition, the risk for irreversible shrinkage is expected to be lower because
wetter conditions will lower the risk of severe drying of the soil. Moreover, soil moisture and temperature are main drivers of
peat decomposition (Boonman et al., 2022). Less deep phreatic groundwater levels in warm summer are expected to lead to
less loss of soil volume due to aerobic peat decomposition by microbial activity, because oxygen can intrude less deep into the
soil creating suboptimal (soil moisture) conditions for peat decomposition. For these reasons, it is expected that WIS reduce
longer term (multiyear) land subsidence. The timeseries presented in this study are not yet sufficiently long to be used to
reliably test these hypotheses. The longest timeseries available from this study, the ~4 years timeseries of Rouveen, does





however support this theory. Based on the extensometer data series of Rouveen, subsidence rates of 4 and 13 mm yr$^{-1}$ for the reference and PWIS parcel respectively were assessed. These rates are in the same order of magnitude as subsidence rates in similar peat areas found in other Dutch (e.g., Schothorst, 1977, Beuving and Van den Akker, 1997, Massop *et al.*, in prep) and international (Bloom, 1964; Haslett *et al*. 1998; Edwards, 2006; Fritz, 2006; Long *et al.*, 2006; Törnqvist *et al.*, 2008; Horton and Shennan, 2009) studies. Variations in both time and space in subsidence rates may be caused by variations thickness of the peat layer, peat type, peat decomposition rate, peat density, intensity of drainage, and climatic conditions (Egglesmann, 1976; Asselen et al, 2009).

The higher subsidence rate in the PWIS parcel of Rouveen is presumably related to (unintendedly) deeper average phreatic groundwater levels in this parcel causing increased peat decomposition and compaction. This observation supports the existence of a positive relation between phreatic groundwater level depth and long-term land subsidence. For the other study locations, longer time series are needed to verify if less deep groundwater levels result in less land subsidence. Also for Rouveen, longer time series are desirable in order to derive more trustworthy subsidence rates. Longer data series allow to filter out short term (seasonal) vertical soil movements that are often an order of magnitude higher than subsidence rates. The effect of groundwater levels on long term subsidence is shown by Massop et al. (submitted), who demonstrate that, based on long-term (circa 50 years) spirit level measurements of surface level and soil markers at different depths, long-term subsidence in a field with relatively high ditch water levels (and related higher phreatic groundwater levels) was less than in a field with relatively low ditch water levels (and related lower phreatic groundwater levels).

**4.2 Spirit levelling versus extensometer technique**

In this study, spirit levelling and extensometery have been used to assess soil movement dynamics and multiyear land subsidence in peat meadow areas. Both field techniques can measure elevation changes at mm-scale precision. An important difference between these two techniques relates to their temporal and spatial applicability. An extensometer measures vertical soil movement at one location, and, therefore, does not provide information on spatial variability of vertical soil movement within a parcel. The temporal resolution is high however (hourly measurements), and soil movement is measured at various depths. This allows detailed analysis of land movement dynamics and subsurface deformation and relate these to other conditions like groundwater level dynamics and variations in subsurface composition. This is vital to understand and unravel different processes contributing to land movement dynamics and long-term subsidence. Spirit levelling has a higher spatial coverage, measuring along multiple transects, but the temporal resolution is commonly low (four times a year in this study, often less), which brings the risk of missing vertical dynamics. Indeed, our measurements demonstrate that the range derived from the high-resolution extensometer measurements is often higher than the range derived from spatially averaged spirit levelling data (see result section). Dynamics derived from extensometer measurements averaged for the day of spirit levelling are, however, often lower compared to dynamics derived from spirit levelling, which may be due to deformation of the top ~5 cm soil between surface level and the topmost extensometer anchor, which is not registered by the extensometer surface anchor.





These differences should be carefully considered and taken into account when using these methods for measuring vertical soil movement.

**5. Conclusions**

The levelling and extensometer data series presented in this study show that yearly vertical soil dynamics, i.e., seasonal subsidence, of the surface is in the order of centimetres. The largest vertical dynamics of almost 10 cm has been measured in an area with a relatively thick (6 m) peat layer, in the dry and warm summer of 2022. Seasonal vertical soil movement is about an order of magnitude higher than long-term subsidence rates in drained peatlands (commonly mm yr$^{-1}$). Therefore, multiyear

data series are needed to filter out seasonal (variable) dynamics and estimate long-term subsidence. In Rouveen, the 4-years extensometer data series allowed to make a first estimate of subsidence rates which were 4 and 13 mm yr$^{-1}$ for the reference and PWIS parcel respectively. Note the (unintended) higher rate derived for the PWIS parcel at this upward seepage location, presumably due to lowered phreatic groundwater levels because of increased drainage of seepage water. For the other study locations with about 2 years of measurements, but preferably also for Rouveen, longer data series are needed to make (more)

reliable estimates of land subsidence.

Results of this study demonstrate that short-term (daily to monthly) vertical soil movements are related phreatic groundwater level fluctuations, which affect reversible deformation processes in both the unsaturated and saturated zone. Water infiltration systems in peat meadows, if correctly applied, limit phreatic groundwater level lowering in summer periods, and therewith,

also limit seasonal subsidence. Less deep summer phreatic groundwater levels cause less shrinkage in the unsaturated zone, due to higher water content and lower suction forces. In the saturated zone, higher phreatic groundwater levels cause less poroelastic deformation because of reduced changes in pore water pressure, and therewith, effective stress. Poroelastic deformation of the saturated subsoil (below ~0.80 m depth) may contribute considerably to surface level movement, often more than processes in the unsaturated zone, especially if the subsurface is composed of a relatively thick peat layer.


Differences in the temporal and spatial resolution of the spirit levelling and extensometer techniques requires careful consideration for developing land movement monitoring plans. Spirit levelling measurements give insight into the spatial variability of elevation and elevation changes within a parcel. This technique is especially useful for long-term (at least 10 years of measurements) application to determine average long-term subsidence of a parcel. Extensometer measurements are

particularly useful for high-resolution monitoring of soil movement dynamics at different depths, which allows to quantify both short and long-term soil movement, and to better understand soil deformation processes in different soil layers and their relative contribution to surface level movement.



## Author contributions

S. van Asselen performed all analyses and prepared the manuscript, with contributions from all authors, and others (see
Acknowledgements).

## Competing interests

The authors declare that they have no conflict of interest.

## Acknowledgements

This research was cofunded by the WUR internal program KB34 Towards a Circular and Climate Neutral Society (2019-2022), project KB34-005-001 (Peat areas in new circular and climate-positive production systems). Furthermore, we would like to thank Harry T.L. Massop, Paul A. Gerritsen, Stefan T.J. Weideveld, Oswin van der Scheer, Enno van Waardenberg, Dennis Peters, Kevin Mouthaan, Sannimari Käärmelahti, Peter Cruijsen, Roy Belderok and Mark Stoetzer for all of their fieldwork related activities. Siem Jansen is thanked for his help with data analysis and visualization.

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
