# Peer review of "Effects of subsurface water infiltration systems on land movement dynamics in Dutch peat meadows"

_Hydrology and Earth System Sciences, 2024_

## Author Comment (AC2)

The manuscript explores the effects of subsurface water infiltration systems (WIS) on vertical soil movement and land subsidence across a collection of peat meadows. The study is data-driven and is based on very detailed experimental campaigns. The sites investigated cover a variety of settings typical of the targeted regional scenario. On one side, I do find the experimental study to be interesting and informative. On the other side, I do think the hydrological component of the study is still not fully developed, the experimental approach and techniques being mostly associated with geotechnical and soil mechanics areas. In this sense, I find the discussion to be focused mostly on the description of the results encapsulated in the figures rather than providing clear interpretations linking fundamental hydrological processes. Hence, I would suggest expanding this element, which is important for the Journal and its readership, and perhaps relegating some more technical and descriptive parts (e.g., local geological/sedimentological settings) to Appendices.

*The focus of this paper is on vertical land movement and driving soil deformation processes, and not on hydrological processes. Hydrology is however an important driver of vertical land movement in peat soils, which is explained in sections 4.1.2 and 4.1.3. The authors think the topic fits very well in HESS, as this is not only about hydrology but also about earth system sciences. For your information, a more in-depth scientific paper on groundwater level measurements at the same study sites is currently being written.*

*Depending on the final review, we would be fine with moving parts of the descriptions of the sites to an appendix. E.g. the maps and the introductory text for each site could be moved. This could for example be replaced by an overview table with some of the main characteristics of the 5 sites. The tables with info on anchors should remain in the main text in our opinion.*

I also found the focus on climate change to support the importance/impact of the study to be too much highlighted and more oriented towards practical applications, rather than uncovering/analyzing fundamental hydrological processes. I would suggest diminishing the emphasis on such element and highlighting more clearly the importance and collocation of the study in the context of the current literature associated with fundamental processes. In essence, the Authors should be clear about whether their contribution is more application-oriented or geared toward providing enhanced understanding of hydrological processes and system functioning. Since I do see a lot of potential in this sense, I would then suggest de-emphasizing the application-oriented aspect.

*Climate change was only mentioned in the introduction. We rephrased the paragraph where it was mentioned, but we still mention climate change because it is one of the reasons this research has been carried out (reduce $CO_2$ emissions from drained / subsiding peat soils. Measures like water infiltration systems are also applied for optimizing water management and reducing land subsidence in times of climate change. We do already make the link between soil deformation processes and their main drivers in the introduction. Moreover, the main focus of this paper is on vertical land movement in peat soils, of which hydrological processes are one of the drivers.*

In terms of quality of results, I did not find too many comments about data uncertainties. For example, I am assuming that groundwater levels are associated with some uncertainties. How are uncertainties associated with all of the data types analyzed impact on potential relationships between processes? Is there a way the Authors can provide some insights on these aspects?

*The spirit levelling and extensomery techniques measure at millimeter scale accuracy. It should be realized that we do present averages of about 100 levelling measuring points. If desired, we could include standard deviations of the mean (standard deviation of changes in soil height relative to the first measurements, otherwise it would be an indication of how irregular the surface is). The uncertainty of the groundwater level measurements is somewhat larger, maybe up to a few centimeters (uncertainty of the device and possible difference between groundwater level in the soil and in the monitoring well). In any case, considering the dynamics of both vertical movement and groundwater level, we expect that these*

*uncertainties will not have any effect on the outcomes and conclusions of the paper. We could include above mentioned notes about uncertainties in the final manuscript.*

Are some of the results (for example, the results depicted in Figure 14) to be expected? If so, is there a rationale underlying such expectation? Or do they come as unexpected? These are some examples of insights that Authors could provide to enhance the potential impact of their work.

*Yes, this is explained in section 4.1.2. We will refer to this section in section 4.1.1.*

Additionally, are these types of results typical of the context they Authors analyze? Or can they be somehow transferred to other settings?

*Yes, this is further explained at the end of section 4.1.2.*

The Authors attempt providing a fit to the data. Why do they expect a linear trend? Is this simply to identify a trend or can this be employed to do something more, e.g., to provide some interpretive model. In any case, when performing a model calibration, I assume the Authors have also evaluated uncertainties associated with parameter estimates. I was not able to see bounds of uncertainty around the plotted linear trends. I would suggest an in-depth analysis of this element together with a clarification of the actual purpose of providing a linear trend line. This is also in line with the statement made by the Authors regarding obtaining an improved quality fit with more data (the Authors refer to a manuscript which is still in the writing phase). Why should the reader be interested to what the Authors define a better fit? How do the Authors quantify the terminology better fit? Simply in terms of R2? Model parameter uncertainty? Model predictive power? The reader would benefit from this kind of discussion, in my view.

*We used linear fits because we want to assess of the average subsidence (rate) for a certain period. We would need a much longer time frame to be able to detect other types of trends. Also, other studies that did have longer time series (a few decades) show that a linear trend is best to use for assessing an average subsidence rate (e.g. Massop et al., 2024, Schothorst, 1977). Moreover, the trendlines give a first indication of the long term subsidence trend (further, the paper focuses on seasonal vertical land movement). Longer time series will give more reliable estimates because yearly differences in land movement dynamics are increasingly filtered out more, and the $R^2$ will increase. We believe an in-depth uncertainty analysis will not add much to the paper at this stage, the time series are still too short. The paper demonstrates methods used and first (promising) results, and is likely to have a follow up paper with more in-depth analyses at a later stage. These lines of reasoning could be highlighted more in the final manuscript.*

When discussing about temporal data series, did the Author observe any trend/drift associated with measurement accuracy? Any induced correlation among data?

*No, this has not been observed*

On these bases, I would suggest a series of revisions that I would define as ranging between moderate and major.

---

## Author Response (AR1)

Subject: revision manuscript hess-2024-152

Dear Serena Ceola,

I have revised the research article HESS-2024-152: Effects of subsurface water infiltration systems on land movement dynamics in Dutch peat meadows. Two reviewers have reviewed the manuscript. The main reviewers' comments highlighted by you were:

1. Provide clear interpretations linking fundamental hydrological processes embedded in the experimental approach here presented;
2. Analyze and comment on data uncertainties, by improving and expanding the results section;
3. Include results of the layers deformation at different depths, as it is useful information to understand and unravel the physical processes leading to the reduction of deformation.

We have revised the manuscript following the suggestions of the reviewers and made by you, which we also have further specified and agreed on in a separate letter (23 December 2024) and email (7 January 2025). Most importantly, we have included additional explanations and figures concerning above-mentioned points. Specifically, we have given more hydrological contextualization in the Results section and also in de Discussion section by describing the relation between two hydrological characteristics (yearly groundwater dynamics and deepest groundwater level in summer) and vertical soil movement. In addition, we included some hydrological characterization in section 2.4 (e.g. about precipitation deficits for the years considered). We also have extended the time series to include 2023, which did not affect our main findings, but amongst others allowed to make better estimates of the long-term average subsidence rates for all locations, and evaluate the effect of WIS on these rates. For details on the revisions see below.

We hope that we have met all comments in an acceptable way.

Best wishes,

Sanneke van Asselen, and co-authors

**Comments reviewer 1, manuscript hess-2024-152**

The work presented by van Asselen et al. investigates the effects of subsurface infiltration systems in subsoil vertical dynamics. Five study sites in the Dutch peat meadows are considered where the installation of extensometers in parcels with and without Water Infiltration Systems (WIS) enables the continuous measurements of the vertical dynamics. Campaigns of spirit leveling are also employed to measure vertical movements of soil and compared to extensometers. The extensometers have different anchor depths to gain deformation of the soil at different levels.

The main outcome of the study is the correlation between water level fluctuations (from a previous study) and surface vertical movements, showing the capability of WIS to increase the phreatic groundwater level thus reducing seasonal land surface fluctuations.

The paper is well-written and well-organized. The concepts are clearly explained as well as the experimental setup.

I think minor revisions are necessary to clarify/improve/discuss the following points:

- As the author stated, there is a discrepancy between the spirit levelling and the extensometers. Dynamics from the extensometers are often lower than the dynamics measured by the spirit levelling. The authors state that this may be due to the deformation of the top 5 cm of soil which is not taken into account in the extensometers. I am not convinced this is the reason. Could be linked to the anchoring system of the extensometers? How is the horizontal plate anchored to the ground?

  *The top anchor is a square perforated stainless-steel plate (0.5 x 0.5 m, 8 mm perforation, about 40% open area) that is dug into the soil at a depth of circa 5 cm (the top 5 cm soil are removed before installation and are put back on top of the plate after installation). A displacement sensor is attached to the plate. This has been done on purpose, to prevent/minimize vertical movement of the plate itself by vegetation growth at the (shallow surface). The setup is carefully designed to measure vertical movement at this depth by soil deformation processes. This has been explained in section 2.2 (last bullet in the list; text has been added to further clarify).*

  *Moreover, lower dynamics obtained from extensometer measurements, as compared to levelling, are only observed when extensometer dynamics are calculated based on averages of measurements on de levelling days (four times a year). When the full extensometer data series are considered, extensometer dynamics are usually higher than dynamics calculated based on levelling. We believe a likely explanation for the observed (structural) differences is deformation caused by shrinkage and swelling of the top 5 cm, as indicated in paragraph 3.6 (sixth bullet in the list, we rephrased this part).*

- The reader is expected to see the results of the layers deformation at different depths. However, the analysis of these data is postponed to a future study. However, I think this is useful information to understand and unravel the physical processes leading to the reduction of deformation with the WIS implementation.

  *New figures have been included showing the deformation of the unsaturated layer between ~5 and 80/115 cm depth and the saturated layer below. The figures have been included in the Appendix, but are referred to in the main text. Text is added about where in the soil profile most soil deformation took place for the different study sites.*

- Is there any way to relate the soil lithology to the high/low result in decreasing the seasonal soil fluctuation? The lithological data available in the study have not been used to interpret the results.

  *This is indeed something we would also like to do, but it is still difficult to do this because there are many factors influencing vertical soil movement. We have however added some (speculative) remarks about this in paragraph 3.6 (fourth bullet).*

- The results from observations of vertical land movements are presented in terms of elevation change. To understand the mechanisms driving the reduction of deformation (poroelastic effect, shrinkage/swelling), I think it could be beneficial to plot the deformations from the surface (-0.05 m) to the -0.8 m anchor. Which is the behavior in time? Example from Fig. 11 (ASD reference but also with AWIS). Why the Anchor 0.06 and 0.79 have an overlapping behavior until March 2022 (no deformation from 0.06 to 0.79)?

  *New figures have been added (see second bullet above). The reason for the overlapping behavior of anchors 0.06 and 0.79 is likely the wet conditions that prevailed in both parcels and the (associated) comparable groundwater level dynamics.*

**Comments reviewer 2, manuscript hess-2024-152**

The manuscript explores the effects of subsurface water infiltration systems (WIS) on vertical soil movement and land subsidence across a collection of peat meadows. The study is data-driven and is based on very detailed experimental campaigns. The sites investigated cover a variety of settings typical of the targeted regional scenario. On one side, I do find the experimental study to be interesting and informative. On the other side, I do think the hydrological component of the study is still not fully developed, the experimental approach and techniques being mostly associated with geotechnical and soil mechanics areas. In this sense, I find the discussion to be focused mostly on the description of the results encapsulated in the figures rather than providing clear interpretations linking fundamental hydrological processes. Hence, I would suggest expanding this element, which is important for the Journal and its readership, and perhaps relegating some more technical and descriptive parts (e.g., local geological/sedimentological settings) to Appendices.

*Following this remark, and after consultation with the editor, we have enhanced the hydrological contextualization. In the Result and Discussion sections, we have included additional hydrological interpretation of the results, especially highlighting more and describing the effect of WIS on groundwater level dynamics per study site (and referring to Van Asselen et al., 2023 in which these effects are explained in detail), and through that the effect on vertical soil movement. In the discussion we describe two hydrological characteristics that affect soil deformation processes, and the effect of WIS on these characteristics. We subsequently relate this to specific soil deformation processes in the unsaturated and saturated soil zone.*

*We have not moved descriptive parts to Appendices, but are also not against it, so are willing to do this if desired.*

I also found the focus on climate change to support the importance/impact of the study to be too much highlighted and more oriented towards practical applications, rather than uncovering/analyzing fundamental hydrological processes. I would suggest diminishing the emphasis on such element and highlighting more clearly the importance and collocation of the study in the context of the current literature associated with fundamental processes. In essence, the Authors should be clear about whether their contribution is more application-oriented or geared toward providing enhanced understanding of hydrological processes and system functioning. Since I do see a lot of potential in this sense, I would then suggest de-emphasizing the application-oriented aspect.

*Climate change was only mentioned in the introduction. We rephrased the paragraph where it was mentioned, but we still mention climate change because it is one of the reasons this research has been carried out (reduce $CO_2$ emissions from drained / subsiding peat soils). Furthermore, we now have enhanced the hydrological contextualization, as mentioned above.*

In terms of quality of results, I did not find too many comments about data uncertainties. For example, I am assuming that groundwater levels are associated with some uncertainties. How are uncertainties associated with all of the data types analyzed impact on potential relationships between processes? Is there a way the Authors can provide some insights on these aspects?

*The spirit levelling and extensomery techniques measure at millimetre scale accuracy, which has now been mentioned in sections 2.1 and 2.2. We also added remarks on that we present spirit levelling averages of about 100 levelling measuring points per field and measurement campaign, and indicate the order of magnitude of the standard deviation of the average elevation changes relative to the first measurement.*

*The uncertainty of the groundwater level measurements is somewhat larger, assessed up to a few centimetres (explained in section 3.6: uncertainty of the device and possible difference between groundwater level in the soil and in the monitoring well). We conclude that, considering the relatively high dynamics of both vertical soil movement and groundwater level, it is unlikely that mentioned uncertainties will change the main findings of the paper.*

Are some of the results (for example, the results depicted in Figure 14) to be expected? If so, is there a rationale underlying such expectation? Or do they come as unexpected? These are some examples of insights that Authors could provide to enhance the potential impact of their work.

*Yes, results are mainly as expected, which is explained in section 4.1.1 and 4.12.*

Additionally, are these types of results typical of the context they Authors analyze? Or can they be somehow transferred to other settings?

*Yes, this is further explained at the end of section 4.1.2.*

The Authors attempt providing a fit to the data. Why do they expect a linear trend? Is this simply to identify a trend or can this be employed to do something more, e.g., to provide some interpretive model. In any case, when performing a model calibration, I assume the Authors have also evaluated uncertainties associated with parameter estimates. I was not able to see bounds of uncertainty around the plotted linear trends. I would suggest an in-depth analysis of this element together with a clarification of the actual purpose of providing a linear trend line. This is also in line with the statement made by the Authors regarding obtaining an improved quality fit with more data (the Authors refer to a manuscript which is still in the writing phase). Why should the reader be interested to what the Authors define a better fit? How do the Authors quantify the terminology better fit? Simply in terms of R2? Model parameter uncertainty? Model predictive power? The reader would benefit from this kind of discussion, in my view.

*We used linear fits because we intend to assess of the average subsidence rate for a specific period. We would need a much longer time frame to be able to detect other types of trends. Also, another subsidence study that did have longer time series (a few decades) for the Zegveld site showed that a linear trend is adequate for assessing an average subsidence rate (Massop et al., 2024). Moreover, we describe that the trendlines give a first indication of the long term subsidence trend. Longer time series will give more reliable estimates because yearly differences in land movement dynamics are increasingly filtered out better, and the $R^2$ will increase. We have now explained this in section 2.4, and give linear fits for all sites in the results section, now we have included also data from 2023. We believe an in-depth uncertainty analysis will not add much to the paper at this stage, because the time series are still relatively short.*

When discussing about temporal data series, did the Author observe any trend/drift associated with measurement accuracy? Any induced correlation among data?

*No, this has not been observed*

On these bases, I would suggest a series of revisions that I would define as ranging between moderate and major.